# Assessing modifications to the Abdul-Razzak and Ghan aerosol activation parameterization (version ARG2000) to improve simulated aerosol–cloud radiative effects in the UK Met Office Unified Model (UM version 13.0)

**Pratapaditya Ghosh[1,2], Katherine J. Evans[3], Daniel P. Grosvenor[4,5], Hyun-Gyu Kang[3], Salil Mahajan[3], Min Xu[3], Wei Zhang[3], and Hamish Gordon[2,6]**

[1]Department of Civil and Environmental Engineering, Carnegie Mellon University,
5000 Forbes Avenue, Pittsburgh, PA 15213, USA
[2]Center for Atmospheric Particle Studies, Carnegie Mellon University, 5000 Forbes Avenue, Pittsburgh, PA 15213, USA
[3]Oak Ridge National Laboratory, Oak Ridge, TN, 37831, USA
[4]Met Office Hadley Centre, Exeter, UK
[5]School of Earth and Environment, University of Leeds, Leeds, UK
[6]Department of Chemical Engineering, Carnegie Mellon University, 5000 Forbes Avenue, Pittsburgh, PA 15213, USA

**Correspondence:** Hamish Gordon (gordon@cmu.edu)

**Abstract.** The representation of aerosol activation is a key source of uncertainty in global composition-climate model simulations of aerosol–cloud interactions. The Abdul-Razzak and Ghan (ARG) activation parameterization is used in several global and regional models that employ modal aerosol microphysics schemes. In this study, we investigate the ability of the ARG parameterization to reproduce simulations with a cloud parcel model and find its performance is sensitive to the geometric standard deviations (widths) of the lognormal aerosol modes. We recommend adjustments to three constant parameters in the ARG equations, which improve the performance of the parameterization for small mode widths and its ability to simulate activation in polluted conditions. For the accumulation mode width of 1.4 used in the Met Office Unified Model (UM), the modifications decrease the mean bias in the activated fraction of aerosols compared to a cloud parcel model from $-6.6\%$ to $+1.2\%$. We implemented the improvements in the UM and compared simulated global cloud droplet concentrations with satellite observations. The simulated cloud radiative effect changes by $-1.43\,\mathrm{W\,m^{-2}}$ (6 %) and aerosol indirect radiative forcing over the industrial period changes by $-0.10\,\mathrm{W\,m^{-2}}$ (10 %).

## 1 Introduction

Climate models usually use parameterizations of aerosol activation to calculate cloud droplet number concentrations ($N_\mathrm{d}$). The Abdul-Razzak and Ghan (2000) activation scheme (hereafter "ARG") is probably the most widely used parameterization. Its popularity is likely due to its simplicity and computational affordability and its sound grounding in the main physical mechanism for activation (adiabatic cooling leading to supersaturation of water). The parameterization is based on detailed comparisons with cloud parcel models. It estimates the maximum supersaturation and, hence, the number of activated aerosols in a grid box using the aerosol size distribution and hygroscopicity, the vertical wind speed (updraft), the ambient temperature, and water surface tension.

Although the ARG scheme is widely used, there are a number of alternative parameterizations, some of which address some assumptions and drawbacks in the ARG scheme. Examples include the Morales Betancourt and Nenes (2014) scheme (hereafter MBN, see also Nenes and Seinfeld, 2003), the Shipway and Abel (2010) parameterization, the Ming et al. (2006) scheme, or the Cohard et al. (1998) scheme. The MBN scheme, for example, splits the entire population

of aerosols into different categories based on whether their size is close to their critical diameter to better estimate the droplet concentrations by accounting, where needed, for kinetic limitations on droplet growth (which the ARG scheme does not). Table 1 of Ghan et al. (2011) lists important differences among several popular activation schemes. While certain schemes demonstrate superior performance to the ARG scheme in many scenarios (Ghan et al., 2011), their integration into existing climate models can be challenging compared to the relatively straightforward implementation of the ARG scheme. Some schemes can be a factor of 20 to 100 more computationally expensive compared to ARG. To mitigate these difficulties, machine learning-based emulators for the calculation of $N_d$ (for example Rothenberg and Wang, 2017; Silva et al., 2021) are emerging. Our premise here, however, is that there are advantages (mainly simplicity and interpretability) to retaining, but updating, existing parameterizations, and in making only minimal changes to climate model code.

The aerosol microphysics schemes adopted in many climate models use three to seven lognormal size modes, and are double-moment in that they prognose both aerosol number and mass concentrations. The widths of the modes are prescribed constants. However, the prescribed widths of the accumulation mode vary substantially between models, between around 1.4 and 2.0 (Table 1), while the width of the Aitken mode is usually around 1.6 or 1.7. Testing of the ARG scheme has been reported by Abdul-Razzak et al. (1998) and Abdul-Razzak and Ghan (2000) focused on the upper end of the range of accumulation mode widths, but Ghan et al. (2011) investigated how the ARG and other activation schemes' performance varies with mode width in their Fig. 6. They used ammonium nitrate aerosols of hygroscopicity 0.7 (lognormally distributed with number geometric mean diameter of 100 nm) and a fixed updraft velocity of $0.5\,\text{m s}^{-1}$ to prepare that figure. Biases in the activation fraction in most schemes were small at widths of around 2.0 but increase below this. At a width of 1.6, the activation fraction in the ARG scheme is $\sim 15\,\%$ lower than the cloud parcel model, however at a width of 2.0 the fractions activated are within 10 %. Nenes and Seinfeld (2003) also showed large deviations of the ARG scheme from their parcel model when $\sigma_{\text{acc}} = 1.5$ (their case SM3 in their Fig. 11).

Clear demonstrations of the impact of neglecting kinetic limitations on droplet activation are provided by Phinney et al. (2003) in their Fig. 1, as well as by Nenes et al. (2001). In reality, as represented by parcel models, $N_d$ increases monotonically as cloud condensation nuclei (CCN) concentrations increase, at least until very large CCN concentrations, usually above $10^4\,\text{cm}^{-3}$, when water uptake by the CCN prevents activation. At this point $N_d$ decreases to zero. However, the uptake of water is kinetically limited, and if this kinetic limit is neglected, activation can be suppressed at concentrations of CCN that are too low. This unrealistic suppression occurs in the ARG parameterization in polluted conditions.

Depending on updraft speed, it can occur when CCN concentrations exceed values as low as $500\,\text{cm}^{-3}$. Giant CCN, if present, may introduce further kinetic limitations due to their large size (Barahona et al., 2010); however, these are not a focus of this current study.

In this work, we revisit the ARG scheme, determine adjustments to two simple tuned functions within it, and test these for accumulation mode widths from 1.4 to 2.1 and over a wide range of other aerosol and environmental parameters, by comparing the performance of the scheme with a cloud parcel model. We also suggest that when kinetic limitations on droplet growth become important, the performance of the scheme can be improved by a simple fix. In Sect. 2, we describe the models we use and the optimization procedure. In Sect. 3, we demonstrate the performance of the modified ARG in the offline parcel model, then evaluate it in the atmosphere component of a global climate model with prognostic double-moment modal aerosol microphysics. Finally, we calculate the impact of the changes (with the fix for kinetic limitations) on the simulated cloud radiative effects and aerosol indirect radiative forcing.

## 2 Methods

Many activation parameterizations, including ARG, were developed and verified against numerical cloud parcel models. We follow these studies to test and improve ARG's performance through comparisons with the open source adiabatic cloud parcel model Pyrcel, developed by Rothenberg and Wang (2016, 2017). The model follows the fundamental equations of Pruppacher and Klett (2010) within the theoretical framework presented by Nenes et al. (2001) using Köhler theory (Kohler, 1936) for droplet formation. The relationships between maximum supersaturation, background aerosol particle growth, and cooling sources are represented using coupled differential equations, as described in Ghan et al. (2011). These equations are then numerically integrated forward over time. The model can accommodate aerosols of varying sizes, concentrations, and hygroscopicities using a number of lognormal distributions. These distributions are approximated as a number of size sections, while numerically solving the differential equations. We use Pyrcel to calculate the maximum supersaturation ($S_{\text{max}}$) and the total activated fraction (TAF) for a given aerosol distribution. We define the TAF as the ratio of the total number of droplets activated to the sum of the number concentrations of aerosols in the Aitken, accumulation, and coarse modes. In these simulations, we do not represent the entrainment of dry air into rising moist parcels.

Kreidenweis et al. (2003) discussed the differences among cloud parcel models: results are influenced by differences in numerical integration methods, variations in the bin sizes for the aerosol and droplet size distributions in size-resolved models, and approaches to evaluate cloud drop activation and

**Table 1.** Geometric standard deviation of Aitken, accumulation, and coarse mode aerosols ($\sigma_{\text{ait}}$, $\sigma_{\text{acc}}$, $\sigma_{\text{coa}}$), used in the microphysics schemes of different climate models are listed. All these models use the ARG activation parameterization to calculate cloud droplet number concentrations.

| Model | $\sigma_{\text{ait}}$ | $\sigma_{\text{acc}}$ | $\sigma_{\text{coa}}$ | Source |
|---|---|---|---|---|
| Unified Model (GLOMAP) | 1.59 | 1.4 | 2.0 | Mann et al. (2010, 2012) |
| CESM2 (MAM4) | 1.6 | 1.8 | 1.8 | Liu et al. (2012, 2016) |
| E3SMv2 (MAM4) | 1.6 | 1.8 | 1.8 | Liu et al. (2012, 2016) |
| ECHAM (HAM2.3) | 1.59 | 1.59 | 2.0 | Stier et al. (2005); Tegen et al. (2019) |
| ICON-A-HAM2.3 | 1.59 | 1.59 | 2.0 | Stier et al. (2005); Salzmann et al. (2022) |
| ECHAM (GMXe) | 1.69 | 1.69 | 2.0/2.2 | Pringle et al. (2010) |
| EMAC (MADE3) | 1.7 | 2.0 | 2.2 | Kaiser et al. (2014) |
| CMAQ (AERO7) | 1.7 | 2.0 | 2.2 | Binkowski and Roselle (2003) |
| MIRAGE | 1.6 | 1.8 | 1.8/2.0 | Easter et al. (2004) |

water activity. We verified that the setup of Pyrcel we use here produces near-identical results to those in Ghan et al. (2011).

In the ARG activation parameterization, the calculation of maximum supersaturation and activated droplet number concentrations relies on two empirical functions, denoted as $f$ and $g$, which depend on the aerosol size mode width $\sigma$. The maximum supersaturation $S_{\text{max}}$ depends on the critical supersaturation $S_m$ of a particular mode and two non-dimensional variables $\eta$ and $\zeta$ defined below. Following Eq. (6) in Abdul-Razzak and Ghan (2000),

$$S_{\text{max}} = \left\{ \sum_i \frac{1}{S_{mi}^2} \left[ f_i \left( \frac{\zeta}{\eta_i} \right)^p + g_i \left( \frac{S_{mi}^2}{\eta_i + 3\zeta} \right)^{3/4} \right] \right\}^{-\frac{1}{2}}, \quad (1)$$

where $i$ indexes aerosol modes in a multi-modal lognormal size distribution and $p = 3/2$. $\zeta$ and $\eta_i$ are given by

$$\zeta = \frac{2A}{3} \left( \frac{\alpha w}{G} \right)^{\frac{1}{2}}$$

$$\eta_i = \frac{2 \left( \frac{\alpha w}{G} \right)^{\frac{3}{2}}}{\gamma^* N_i}. \quad (2)$$

Here $A$ is the Kelvin coefficient (a function of temperature), $w$ is the updraft velocity, $\alpha$ is a thermodynamic term (function of temperature) that relates the updrafts to the tendency for water vapor to condense as it cools, $N_i$ is the aerosol number concentration in mode $i$, $\gamma^*$ follows from the thermodynamics of rising moist air with assumptions listed in Pruppacher and Klett (2010), and $G$ is the growth coefficient which depends on the diffusivity of water vapor in air and on the thermal conductivity of the air. The ratio $\zeta/\eta_i$ is proportional to the ratio $N_i/w$. Since none of these parameters depend of the mode width, Abdul-Razzak and Ghan (2000) parameterized the dependence of $S_{\text{max}}$ on $\sigma$ by using two additional parameters, $f$ and $g$. For each aerosol mode, $f$ and $g$ were determined by comparison to a parcel model (Abdul-Razzak and Ghan, 2000) as

$$f_i = 0.5 \exp \left( 2.5 \ln^2 \sigma_i \right)$$

$$g_i = 1 + 0.25 \ln \sigma_i. \quad (3)$$

These two constants appear in the denominator of the overall equation for $S_{\text{max}}$. Therefore, lower values of $f$ and $g$ would lead to higher $S_{\text{max}}$ (and higher $N_d$). In this paper we propose modifications to $f$, $g$, and $p$.

To calculate improved values of $f$ and $g$ and a value for $p$, we minimized a customized error function over a large number of test cases. We generated 400 Pyrcel simulations spanning the parameter space shown in Table 2 above the horizontal line (i.e., excluding updraft speeds and accumulation mode width) using Latin hypercube sampling (Mckay et al., 2000; Helton and Davis, 2003). We used the PyDOE Python library (Baudin, 2015) with a random seed set to 400. We repeated the Latin hypercube for 25 updraft speeds distributed evenly in logarithmic space from $10^{-3}$ to $10 \, \text{m s}^{-1}$. The lower end represents a lower limit, for updrafts in fog for example, and the upper end represents updrafts in warm convective clouds. We repeated this set of $25 \times 400$ simulations for four different mode widths of the accumulation mode: 1.4, 1.6, 1.8, and 2.1. These values are frequently used in different global climate models (Table 1). Within the Latin hypercubes, the aerosol number concentrations were sampled in logarithmic space, while the other parameters were sampled in linear space. We kept the value of the mass accommodation coefficient fixed at 1.0. We vary only the mode width of the accumulation mode, as we find that it is more important to the determination of $N_d$ than the Aitken and coarse mode widths.

Inspired by the population splitting approach of Nenes and Seinfeld (2003), we divided the 10 000 Pyrcel simulations for each $\sigma$ into two sets according to whether kinetic limitations are or are not important. To identify an analytic criterion to partition the simulations that is compatible with the ARG parameterization, we noted that the $S_{\text{max}}$ predicted by ARG sharply decreases above a threshold $\zeta/\eta_i > 1$, deviating from the $S_{\text{max}}$ predicted by Pyrcel. Given that the ratio $\zeta/\eta_i$ is pro-

**Table 2.** Set of parameters used in different Pyrcel model simulations. Here, "ait", "acc", and "coa" subscripts refer to Aitken, accumulation, and coarse mode aerosols, respectively. $N$ represents number concentration, $D$ geometric mean diameter, $\sigma$ the width of the aerosol size distribution, $w$ updraft speed, and $\kappa$ hygroscopicity. "Yes" or "No" in the LHS column represent whether or not the Latin hypercube sampling technique was used to vary that parameter in a particular range.

| Parameters | Values | LHS |
|---|---|---|
| Pressure (Pa) | [50 662.5, 101 325] | Yes |
| Temperature (K) | [240, 300] | Yes |
| $\kappa$ | [0.1, 1.2] | Yes |
| $N_{ait}$ (cm$^{-3}$) | [10, 10 000] | Yes |
| $D_{ait}$ (nm) | [20, 90] | Yes |
| $N_{acc}$ (cm$^{-3}$) | [10, 10 000] | Yes |
| $D_{acc}$ (nm) | [100, 400] | Yes |
| $N_{coa}$ (cm$^{-3}$) | [0.1, 10] | Yes |
| $D_{coa}$ (nm) | [500, 2000] | Yes |
| $\sigma_{ait}$ | 1.59 | No |
| $\sigma_{acc}$ | 1.4, 1.6, 1.8, 2.1 | No |
| $\sigma_{coa}$ | 2.0 | No |
| $w$ (m s$^{-1}$) | [$10^{-3}$–10]: 25 values (log space) | No |

portional to $N_i/w$, kinetic limitations are more important at high values of $\zeta/\eta_i$, where the supersaturation is lower and therefore aerosols activate more slowly than at lower values of $\zeta/\eta_i$. The threshold of 1 was determined by examining the behavior of Eq. (1); we do not have a clear theoretical justification for the precise value. However, we find it matches the criterion for population splitting in Nenes and Seinfeld (2003) quite well in the cases we examined (Sect. 3.1).

To determine improved values of $f$ and $g$, we calculated the TAF predicted by the ARG scheme for each Pyrcel simulation in the set of simulations for which kinetic limitations are unimportant. We then determined a weighted mean squared error (WMSE) in the total activated fraction in the ARG scheme as defined below:

Weighted mean squared error $=$

$$\frac{1}{\mathcal{N}}\sum_{i=1}^{\mathcal{N}}\begin{cases}\alpha \cdot (\text{ARG}_i - \text{Pyrcel}_i)^2 & \text{if ARG}_i > \text{Pyrcel}_i \\ (\text{ARG}_i - \text{Pyrcel}_i)^2 & \text{otherwise}\end{cases},$$
(4)

where

– Pyrcel$_i$ is the Pyrcel model activated fraction,

– ARG$_i$ is the ARG scheme activated fraction,

– $\alpha$ is an asymmetry factor, and

– $\mathcal{N}$ is the number of samples.

We searched a grid of possible $f$ and $g$ values, spaced by 0.01, for each mode width to find the values that minimize

WMSE. We set the asymmetry factor $\alpha$ to 5, so we severely penalize overestimations of Pyrcel by ARG. We found that without this constraint ($\alpha = 1$), the optimal $f$ has an extremely low value (a factor of 10 times lower than that suggested by ARG), which may not be physical. We also prefer not to overestimate Pyrcel given that the default ARG underestimates it, as overestimating Pyrcel would lead to us overestimating the impact of the modifications on the atmosphere. Finally, we fitted the values of $f$ and $g$ we determined for each accumulation mode width to one-dimensional functions.

Using the improved $f$ and $g$ in the set of simulations for which the kinetic limit is important, we then calculated the activated fractions predicted by ARG for these kinetically limited cases and used them to search for the $p$ values that minimize WMSE. Here, we set the asymmetry factor $\alpha$ to 0.2, severely penalizing underestimations. The use of $\alpha$ here is motivated by the fact that for these cases, we find that the default ARG scheme severely underestimates the TAF in relative terms (in other words, in logarithmic space), often by up to four orders of magnitude. Penalizing underestimations more heavily than overestimations produces good results when the TAF is low, below 0.1, and is evaluated in logarithmic space. We compared the resulting parameterization with MBN as well as the parcel model, for context and as an additional verification of the setup.

To understand how the proposed improvements to the ARG scheme impact global $N_d$, cloud radiative effects, and aerosol indirect radiative forcing, we implemented them in the UK Met Office Unified Model (UM). We used UM version 13.0 in an atmosphere-only configuration of the UK Earth System Model, v1.1 (Mulcahy et al., 2023) with fixed sea surface temperatures. Horizontal winds above the boundary layer were nudged to ERA5 reanalysis. The choice of the nudging relaxation parameter is important; too small a value makes nudging ineffective, while too large a value can destabilize the model. Following Telford et al. (2008), we use the relaxation parameter of $\frac{1}{6}$ h$^{-1}$, corresponding to the time spacing of the ERA5 data. The model configuration is based on GA7.1 (Walters et al., 2019). We simulated the year 2014, with 4 months of model spin-up, with and without modifications to the ARG parameterization. We also ran two additional 1 year long simulations of 2014 with pre-industrial aerosol and aerosol precursor emissions (year 1850) (but no other changes) to estimate changes in radiative forcing. Climate models that participated in the Coupled Model Intercomparison Project Phase 6 (CMIP6) also reported radiative forcing due to aerosol–cloud interactions over a period ending in the year 2014 (Forster et al., 2021). The model setup we use has a horizontal resolution of $1.87° \times 1.25°$ (labeled N96 within the UM framework) with 85 vertical levels in total and 50 levels between 0 and 18 km.

Within the UK Chemistry and Aerosol (UKCA) scheme in the UM, the double moment modal Global Model of Aerosol Processes (GLOMAP-mode, hereafter referred to as

GLOMAP) aerosol microphysics scheme simulates aerosol mass and number concentrations, excluding dust, in 5 log-normal modes (Mann et al., 2010). The improvements of Mann et al. (2012) and Mulcahy et al. (2020, 2023) are included. Dust is represented in a separate scheme and does not participate in activation. Building on the configuration of UKCA used by Mulcahy et al. (2023), we included the boundary layer nucleation scheme of Metzger et al. (2010) following Ranjithkumar et al. (2021) in all simulations, to obtain better agreement of aerosol number concentrations with observations. The CMIP6 inventories (Feng et al., 2020) were used for aerosol and gas emissions.

In GLOMAP, a width $\sigma_{\mathrm{acc}} = 1.4$ is used for aerosols in the accumulation mode (Mann et al., 2012) and $\sigma_{\mathrm{ait}} = 1.59$ for the Aitken mode (Mann et al., 2010). Mann et al. (2012) discussed this choice of a relatively low accumulation mode width compared to other models (see Table 1) in Sect. 7 of their article. They concluded that using a lower width results in better agreement with a sectional microphysics scheme and is supported by some, but not all, observations in the literature. Because the accumulation mode width in GLOMAP is the lowest of all the models we tabulate, we expect to see the largest improvement in this model when we adjust the ARG parameterization.

The model configuration includes the single-moment cloud microphysics scheme of Wilson and Ballard (1999). The $N_{\mathrm{d}}$ at the cloud base are calculated using the UKCA-Activate scheme (West et al., 2014), which uses the ARG activation parameterization. In each grid box, a probability density function (PDF) of updrafts, centered around the resolved updraft speed, with a distribution width set to the minimum of $0.01\,\mathrm{m\,s^{-1}}$ and the square root of two thirds of the turbulent kinetic energy (TKE), is passed to the activation scheme. The activation scheme calculates the expected number of droplets at each time step, diagnostically, for each updraft speed, and then a weighted average of the predicted $N_{\mathrm{d}}$ over positive updraft speeds is computed.

## 3 Results

### 3.1 ARG evaluation

We recommend updated equations for use in the ARG parameterization:

$$f = 0.0135 e^{2.367\sigma_{\mathrm{acc}}}$$
$$g = 1.1058 - 0.315\sigma_{\mathrm{acc}}$$
$$p = \begin{cases} -0.5073 + 1.5088\sigma_{\mathrm{acc}} - 0.3699\sigma_{\mathrm{acc}}^2 \\ \qquad\qquad\qquad\qquad \text{if } \zeta/\eta_i > 1 \\ 1.5 \qquad\qquad\qquad\quad \text{if } \zeta/\eta_i \leq 1, \end{cases} \quad (5)$$

where the subscript $i$ refers to the aerosol size mode.

Unlike the default scheme, here we recommend using the same set of $f$, $g$, and $p$ for all the modes. These parameters

are functions of the accumulation mode width only. The $f$ and $g$ values we obtained for each accumulation mode width $\sigma_{\mathrm{acc}}$ are listed together with the default ARG values (for the accumulation mode) in Table 3. We do not expect the functions in Eq. (5) to be valid outside $1.4 \leq \sigma_{\mathrm{acc}} \leq 2.1$.

We compare the updated and default ARG schemes with the MBN parameterization and the parcel model in Fig. 1. We show $S_{\mathrm{max}}$ and the TAF as a function of updraft speed, for $\sigma_{\mathrm{acc}} = 1.4$ and 2.1. Consistent with the analysis by Ghan et al. (2011), the default ARG scheme (orange lines) underpredicts the supersaturation and TAF almost everywhere and the underprediction is usually more severe for $\sigma_{\mathrm{acc}} = 1.4$ than for $\sigma_{\mathrm{acc}} = 2.1$ (compare subfigures b, j, and n with h, l, and p). Despite some discrepancies, visible in Fig. 1j for example, when we consider all the simulations, the more complex MBN scheme (blue line) is generally in very good agreement with the parcel model across the whole parameter space. The updated ARG scheme (red line in the figures) also shows very good agreement of TAF and $S_{\mathrm{max}}$ with the parcel model. Mean biases in $S_{\mathrm{max}}$ and TAF are given in Table 4 and the sensitivity of TAF to $f$ and $g$ is shown in Fig. S1 in the Supplement. Overall, for $\sigma_{\mathrm{acc}} = 1.4$ the mean bias in TAF was $-6.6\,\%$ before the update and $1.2\,\%$ after. The performance is similar to and often better than the MBN scheme, for example, for updrafts in the range of $0.01–0.5\,\mathrm{m\,s^{-1}}$, as in Fig. 1n. Figure S2 in the Supplement shows the weighted mean squared error in the activated fraction for optimizations performed with asymmetry factors $\alpha = 5.0$ and 1.0, as a function of $f$ and $g$, for $\sigma_{\mathrm{acc}} = 1.4$. $\alpha = 1.0$ is equivalent to the standard definition of the mean squared error. While the minimum WMSE is lower when $\alpha = 1.0$, the corresponding $f$ values are extremely low and so using them would represent a substantial change to the physics of the ARG parameterization, which is not our aim (the $f$ term in Eq. 1 would often become negligible). Figure S2 also shows how $f$ and $g$ are correlated. The default ARG values have a significantly higher error than the values we obtain, as expected from the mean biases in Table 4.

Figure S3 in the Supplement shows how the adjusted ARG scheme performs for alternative combinations of accumulation mode aerosol number concentration and geometric mean diameter. Figure S4 in the Supplement demonstrates how the TAF changes as a function of hygroscopicity, ambient temperature, and pressure while other parameters are kept fixed, i.e., at a particular location in the parameter space. The default ARG scheme underestimates the TAF by around 0.1–0.2 for the cases we tested in Fig. S4, while the updated ARG and MBN schemes are in excellent agreement with the Pyrcel model.

Estimates of $N_{\mathrm{d}}$ were found to be sensitive to variation of the mass accommodation coefficient ($\alpha_c$) by Laaksonen et al. (2005) (see also Raatikainen et al., 2013). In Fig. S5 in the Supplement, we demonstrate that the results we show are also sensitive to $\alpha_c$, which is set to 1 in all other tests. However, for $\alpha_c = 0.1$, the underestimate of TAF and $S_{\mathrm{max}}$ by the

**Table 3.** Recommended values of $f$, $g$, and $p$ for different values of accumulation mode geometric standard deviation ($\sigma_{acc}$).

| $\sigma_{acc}$ | Old $f_{acc}$ | New $f$ | Old $g_{acc}$ | New $g$ | Old $p$ | New $p$ (for $\zeta/\eta_i > 1$) |
|---|---|---|---|---|---|---|
| 1.4 | 0.66 | 0.37 | 1.08 | 0.67 | 1.50 | 0.88 |
| 1.6 | 0.86 | 0.60 | 1.11 | 0.60 | 1.50 | 0.96 |
| 1.8 | 1.18 | 0.53 TS1 | 1.14 | 0.53 | 1.50 | 1.01 |
| 2.1 | 1.97 | 0.45 TS2 | 1.18 | 0.45 | 1.50 | 1.03 |

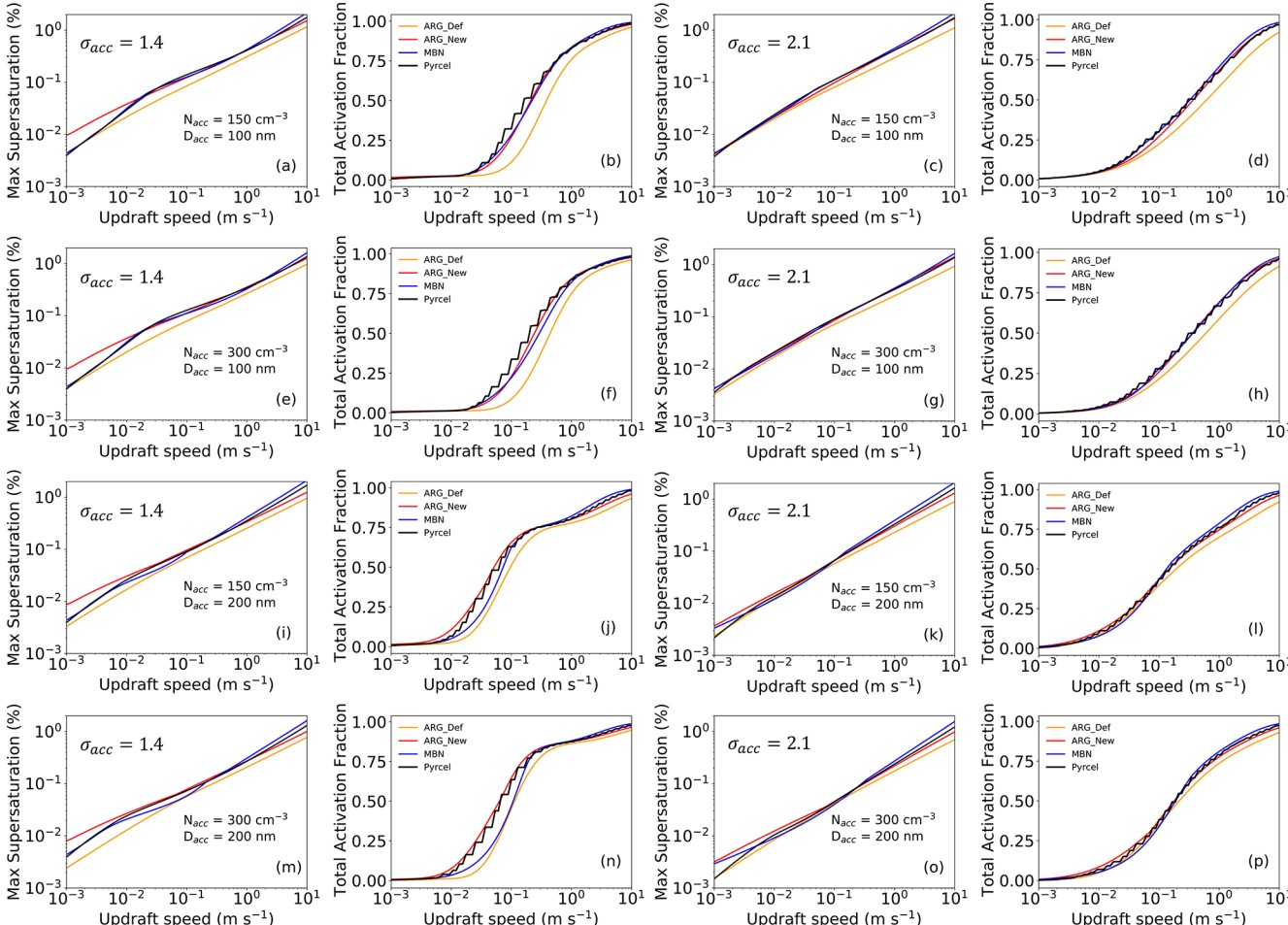

**Figure 1.** Maximum supersaturation ($S_{max}$) and total activated fraction (TAF) predicted by Pyrcel, ARG with and without modifications (ARG_New and ARG_Def in the figure legends), and the MBN parameterization, as a function of updraft speed. The accumulation mode width $\sigma_{acc} = 1.4$ (left side) and 2.1 (right side). The accumulation mode geometric mean diameter is 100 nm (top 4 panels) and 200 nm (bottom 4 panels). The number concentrations are set to 150 cm$^{-3}$ (first and third row) and 300 cm$^{-3}$ (second and fourth row). The Aitken and coarse mode concentrations and geometric mean diameters are kept fixed at 50 cm$^{-3}$, 5 cm$^{-3}$, 40 nm, and 800 nm respectively, while the hygroscopicity is 0.6, the pressure 101 325 Pa, and the temperature 293 K.

default scheme can still be mitigated by using the updates we propose in this work. The bias in the default ARG scheme for the conditions plotted in Fig. S5 changes from $-7.7\%$ to $-5.8\%$ as $\alpha_c$ changes from 1.0 to 0.1, while the bias of the updated parameterization changes from $-1.5\%$ to $0.9\%$.

In Fig. 2 we show $S_{max}$ and total droplet number concentrations ($N_d$) as a function of aerosol number concentrations

in the accumulation mode (up to $10^5$ cm$^{-3}$) for $\sigma_{acc} = 1.4$. The eight different pairs of subfigures show updraft velocities of 0.2 and 1 m s$^{-1}$, Aitken mode number concentrations of 500 and 50 cm$^{-3}$, and accumulation mode diameters of 100 and 200 nm. We compare the default ARG (orange solid lines), the updated ARG (red solid lines), and the MBN scheme (blue solid lines) with the Pyrcel model (black solid

**Table 4.** Mean bias in $S_{max}$ and total activated fraction (TAF) for default and updated ARG, along with the MBN scheme, is listed for different values of accumulation mode sigma. These values are obtained from the Pyrcel model simulations. Both the quantities are unitless and represented as percentages, which we define as mean bias percentage $= \frac{\sum (\text{TAF or } S_{max} \text{ in ARG or MBN} - \text{TAF or } S_{max} \text{ in Pyrcel})}{\text{total number of simulations}} \times 100$. These numbers are calculated for the full parameter space tested in this study.

| Scheme | $\sigma_{acc} = 1.4$ | $\sigma_{acc} = 1.6$ | $\sigma_{acc} = 1.8$ | $\sigma_{acc} = 2.1$ |
|---|---|---|---|---|
| ARG_Def, $S_{max}$ | −0.034 | −0.034 | −0.033 | −0.033 |
| ARG_New, $S_{max}$ | −0.014 | −0.013 | −0.013 | −0.014 |
| MBN, $S_{max}$ | 0.016 | 0.017 | 0.021 | 0.071 |
| ARG_Def, TAF | −6.630 | −5.374 | −4.451 | −3.547 |
| ARG_New, TAF | 1.214 | 1.221 | 1.139 | 0.870 |
| MBN, TAF | −0.221 | 0.672 | 1.333 | 3.234 |

lines). The transitions to the kinetically limited regime in the MBN parameterization are visible as kinks in the maximum supersaturation subfigures. We also show the performance of the model with updated values of $f$ and $g$, but without modifying the power $p$ (green dotted lines), to demonstrate the importance of optimizing the constant $p$ to improve the ARG scheme when kinetic limitations to water uptake are important. For $N_{acc}/w > 10^4 \, \text{cm}^{-3} \, \text{sm}^{-1}$ with the new $f$ and $g$, both the default and the updated parameterizations (with the default $p$) predict a decrease in droplet number concentration with increasing aerosol number concentration, in contrast to the parcel model, as noted by Phinney et al. (2003). The empirical modification to the constant $p$ does not fix the underlying problem, but it does lead to good agreement of the modified ARG parameterization with the parcel model up to around $N_i/w > 10^5 \, \text{cm}^{-3} \, \text{sm}^{-1}$ with the new $f$ and $g$, an order of magnitude higher than before. Unlike the modification we propose here, the approaches of Nenes and Seinfeld (2003); Ming et al. (2006) have physically motivated treatments of kinetic droplet growth. However, introducing additional physics into the ARG scheme would likely lead to a formulation similar to those algorithms, compromising the simplicity. Alternatively, of course, one could also guarantee fidelity to the parcel model using machine learning-based emulators, for example, those of Silva et al. (2021). This would require larger changes to the code base of a climate model to implement and may reduce the ease with which the behavior of the scheme can be interpreted.

The values of $f$, $g$, and $p$ we recommend are independent of the Aitken and coarse mode widths and are a function of accumulation mode width. In Fig. S6 in the Supplement, keeping the accumulation mode width fixed at 1.4, we change the Aitken and coarse mode widths to 1.7 and 1.8. We show that for different cases with these widths, the updated ARG parameterization is still in good agreement with the parcel model.

## 3.2 Effects on cloud droplet number concentrations

In Fig. 3 we compare daytime liquid cloud-top $N_d$ simulated in the UM with observations from the MODerate Imaging Spectroradiometer (MODIS) satellite (Grosvenor et al., 2018). The cloud top is defined as the highest model level with non-zero liquid cloud fraction. If the cloud fraction is 100 %, then $N_d$ at that level is taken as the cloud top $N_d$. If not, the model-level-weighted $N_d$ is calculated by summing over contributions from different model levels here indexed by $i$, as

$$N_d = \frac{\sum_{i=1}^{n} N_{d,i} \cdot F_{lc,i} \cdot P_i}{\sum_{i=1}^{n} F_{lc,i} \cdot P_i}. \tag{6}$$

Here $N_{d,i}$ is the droplet number concentration at the $i$th model level. $F_{lc,i}$ is the liquid cloud fraction at the $i$th model level. $P_i$ is the probability that a photon leaving the $i$th layer can escape to space without encountering a cloud in the layers above. This is calculated based on the cloud fraction in the layer above the $i$th layer. $n$ is the total number of model levels. This equation is calculated only for the daytime. The cloud-top $N_d$ is calculated in each time step, and the monthly average is written out as an output diagnostic. Figure 3 shows annual averages of the difference between the model and observations. We calculated several metrics to quantify the model performance. The normalized mean bias (NMB) is useful in that it describes whether the model overestimates or underestimates observations overall, but positive and negative biases tend to cancel in the metric. Therefore, root mean square error (RMSE) is also commonly used to summarize the magnitude of the differences between the model and observations. However, both metrics penalize fractional overestimates more than fractional underestimates: for example, a model that overestimates observations by a factor two has a NMB of $+100\,\%$, while one that underestimates observations by a factor two has a NMB of $-50\,\%$. The model that overestimates has a larger absolute error but the same fractional error compared to the model that underestimates. Hence, following Gustafson and Yu (2012), we also compute the nor-

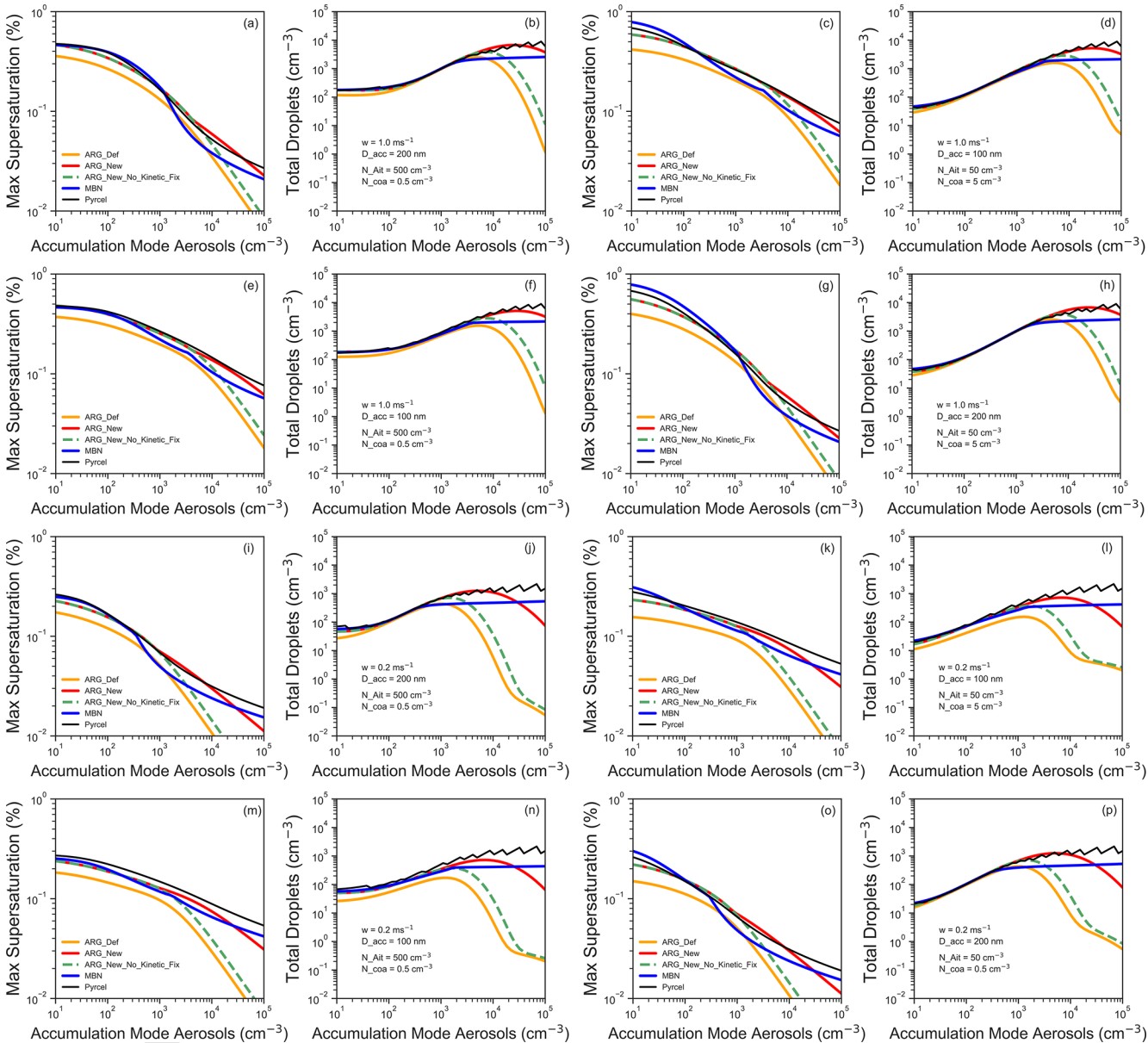

**Figure 2.** Maximum supersaturation and the total droplet number concentration (cm$^{-3}$) as a function of accumulation mode ($\sigma_{\mathrm{acc}} = 1.4$) number aerosol number concentrations ($10$–$10^5$ cm$^{-3}$) for the parcel model, original ARG, updated ARG (ARG_New), updated ARG without the changes to $p$ to account for kinetic limitations on water uptake (ARG_New_No_Kinetic_Fix), and the MBN activation scheme. We used different updraft speeds ($w$), diameter of accumulation mode aerosols (D_acc), and number concentrations in the Aitken and coarse mode (N_ Ait, N_ coa). Geometric mean diameters of Aitken, accumulation, and coarse mode aerosols are kept fixed at 40, 100, and 800 nm respectively. The temperature, pressure, and hygroscopicity are fixed at 293 K, 101 325 Pa, and 0.6.

malized mean absolute error factor (NMAEF), which penalizes fractional overestimates and underestimates equally. It also differs from RMSE in that RMSE depends quadratically on errors while NMAEF depends on them linearly. We quote a simplified version of the NMAEF valid for positive model values and observations; the full version is given in Eq. (5)

of Gustafson and Yu (2012). These metrics are calculated as

$$\mathrm{NMB} = \frac{\sum(\mathrm{model} - \mathrm{MODIS})}{\sum \mathrm{MODIS}}$$

$$\mathrm{RMSE} = \sqrt{\frac{\sum(\mathrm{model} - \mathrm{MODIS})^2}{N}}$$

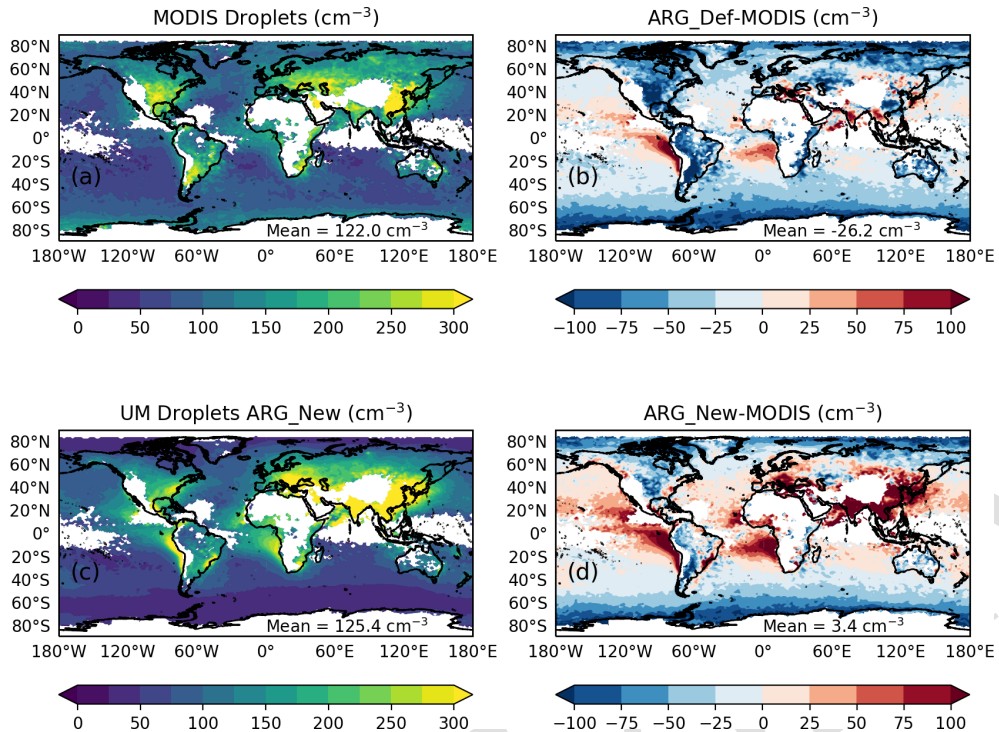

**Figure 3. (a)** Annual average of daytime top of cloud droplet number concentrations from MODerate Imaging Spectroradiometer (MODIS) satellite data (Grosvenor et al., 2018) regridded to the N96 UM grid. **(b)** Bias in the UM simulation with default ARG (ARG_Def). **(c)** Annual average of daytime top-of-cloud droplet number concentrations from UM simulation with updated ARG (ARG_New). **(d)** Bias in UM simulation with updated ARG.

$$\mathrm{NMAEF} = \begin{cases} \frac{\sum |\mathrm{model} - \mathrm{MODIS}|}{\sum \mathrm{MODIS}}, & \text{if } \overline{\mathrm{model}} \geq \overline{\mathrm{MODIS}} \\ \frac{\sum |\mathrm{model} - \mathrm{MODIS}|}{\sum \mathrm{model}}, & \text{if } \overline{\mathrm{model}} < \overline{\mathrm{MODIS}}, \end{cases}$$

where the sum is over all $N$ model grid boxes which have valid MODIS retrievals. To produce these results, we regridded MODIS satellite data to the model grid and calculated monthly average MODIS $N_d$ using the grid points where at least 1 d of observations is available. We filter the model output based on the availability of MODIS retrievals in a specific grid box for each respective month. Although comparing monthly means rather than temporally co-located datasets has representativeness uncertainty, it allows the results to be compared with other studies such as that of Mulcahy et al. (2023). The $N_d$ datasets discussed by Grosvenor et al. (2018) were also compared monthly. Poleward of 30° latitude, in the cloudiest regions (such as the coast of the Pacific North-western United States, the coast of western Europe, or New-foundland in the northwestern Atlantic, for example) which most affect the global average, the updated ARG parameterization leads to an improvement in the NMB from −25.8 % to −5.2 %, while between −30° to 30° latitude, the updates worsen the NMB from −3.2 % to 32.6 %. The globally averaged NMB changes from −21 % to 2.8 %. Figure S7 in the Supplement shows the mean bias as a histogram for default and updated ARG cases. The globally averaged RMSE

increases from 64 to 72 cm$^{-3}$ when the ARG scheme is updated, while the global average NMAEF decreases from 0.48 to 0.39. Thus, the overall bias and fractional error decrease when we update the ARG parameterization, while the absolute error (measured by the RMSE) increases. Whether or not the performance of the model improves overall therefore depends on the priorities of the user. The magnitude of these biases and associated cloud radiative effects might also depend on choice of model (Smith et al., 2020), or version, or the simulation year.

The Grosvenor et al. (2018) dataset has several assumptions and thresholds to calculate $N_d$. An important assumption is a minimum of 80 % cloud cover in a $1° \times 1°$ grid box. The model diagnostics are written out as monthly averages, to avoid the high cost in disk space associated with output at higher time resolution, and hence it is difficult to apply exactly the same threshold. To understand the impact of this cloud fraction threshold on the temporally averaged $N_d$, we design a sensitivity experiment by filtering out the bottom 25th percentile of weights within each month (the denominator in Eq. 6) and their corresponding simulated $N_d$ values, and mask out the same grid cells from the monthly-averaged MODIS dataset. This filtering ensures that we select grid boxes which have relative higher cloud fractions on average in the model. In Fig. S8 in the Supplement, we show

the simulated and observed $N_d$ with this filter in place. We find that the global NMB change with the updated ARG is similar to the unfiltered case (a change from $-22\%$ to $0.1\%$ compared to the previous change from $-21\%$ to $2.8\%$). The changes to the RMSE and NMAEF with the filter are also small (RMSE increases from 59 to $65\,\mathrm{cm}^{-3}$ compared to the previous change from 64 to $72\,\mathrm{cm}^{-3}$). Although this alternative methodology remains approximate, it illustrates the sensitivity of the cloud fraction threshold to $N_d$. However, we did not observe significant changes in the results. Therefore there is no strong evidence from this test that the results would be substantially changed if we were able to match the simulations to satellite data more precisely.

Similarly sized or even larger biases are present in (at least) several other major climate models (for example Saponaro et al., 2020; Christensen et al., 2023). The spatial pattern of the $N_d$ biases (Fig. 3) are consistent with similar evaluations by Mulcahy et al. (2020) and Mulcahy et al. (2023) for the UK Earth System Model. The biases quite likely originate from aerosol modeling, cloud processes or updraft speeds, unrelated to the activation parameterization. The satellite retrievals used for the evaluation are also uncertain, and the comparison is imprecise due to the representativeness uncertainty. However, it is still helpful to understand the impact of the updated functions on how the climate model represents $N_d$ globally, in the context of a roughly estimated bias with respect to observations.

Figures S9 and S10 in the Supplement show plots similar to Fig. 3 for the December–January–February average and the June–July–August average, respectively. The globally averaged mean bias reduces when ARG is updated during both December–January–February and June–July–August. Figure S11 in the Supplement shows the effect of the adjustment of the power $p$ in Eq. (1) to avoid unduly rapid decreases in $S_{\mathrm{max}}$ under polluted conditions. Without this adjustment, the modified ARG scheme predicts considerably fewer droplets in East and South Asia and has a slightly better agreement with MODIS observations, but then the satellite retrievals are also very uncertain in these areas. More detailed study of these polluted conditions is needed.

In Fig. S12 in the Supplement, we present the difference in liquid water path (LWP) between present-day simulations using the default ARG and the updated ARG. The results indicate a small yet systematic increase of $1.72\,\mathrm{g\,m}^{-2}$ ($3.4\%$) in LWP with the updated ARG, concentrated in regions where cloud cover is high. We find only minor differences in precipitation and cloud cover between these simulations (not shown).

## 3.3 Changes in radiative effects

By changing $N_d$ and therefore cloud albedo and life cycle, the updated ARG changes the simulated radiative effects of clouds and aerosols and the aerosol indirect radiative forcing. Changes to the direct aerosol radiative effect,

the cloud radiative effect, and total radiative effect due to aerosol and cloud adjustments ($\Delta$DRE, $\Delta$CRE, $\Delta$TRE respectively) that result from the updates to ARG are shown in Fig. 4. These changes are calculated consistently with the technique for aerosol radiative forcing prescribed by Ghan (2013), though we assume that only changes in DRE and CRE are non-negligible (we ignore aerosol-induced changes to surface albedo, for example). We find that the global mean $\Delta$DRE $= +0.05\,\mathrm{W\,m}^{-2}$, $\Delta$CRE $= -1.43\,\mathrm{W\,m}^{-2}$, and $\Delta$TRE $= -1.37\,\mathrm{W\,m}^{-2}$. Changes in DRE are not locally negligible, most likely because changing cloud albedo below aerosols affects the amount of light the aerosols scatter or absorb (Chand et al., 2009), but also possibly because changing $N_d$ affects aerosol scavenging rates (in the simulations we show, this is due to the Albrecht effect) and therefore affects concentrations of scattering and absorbing aerosol. $\Delta$TRE is dominated by shortwave radiative effects and the contribution of longwave radiative effects is negligible. Over the tropics, $\Delta$TRE is larger, as incident radiative fluxes are higher. The changes in cloud radiative effects that we see are comparable to, but somewhat higher than, those found by Rothenberg et al. (2018), who also found a spread of around $0.8\,\mathrm{W\,m}^{-2}$ when several different activation schemes were included in the same model.

We calculate the approximate changes in radiative forcing (RF) using the simulation with pre-industrial aerosol emissions described earlier. Bellouin et al. (2020) estimates the present-day aerosol effective radiative forcing to range from $-1.60$ to $-0.65\,\mathrm{W\,m}^{-2}$, with a $16\%$–$84\%$ uncertainty range. The Sixth Assessment Report (AR6) of the Working Group I (WGI) of the Intergovernmental Panel on Climate Change (IPCC) estimates the effective radiative forcing due to aerosol–radiation interaction at $-0.3$ ($-0.6$ to $0.0$) $\mathrm{W\,m}^{-2}$ and the effective radiative forcing due to aerosol–cloud interaction at $-1.0$ ($-1.7$ to $-0.3$) $\mathrm{W\,m}^{-2}$ (Forster et al., 2021) over the industrial era (1750–2014). The nudged simulations may not capture all adjustments (Zhang et al., 2014), and hence do not use the term "Effective Radiative Forcing", but the simulated radiative forcings due to aerosol–radiation and aerosol–cloud interactions are reasonable at $-0.20$ and $-0.95\,\mathrm{W\,m}^{-2}$ respectively. We find a mean change in the total RF estimate of $\Delta$TRF $= -0.08\,\mathrm{W\,m}^{-2}$, while the change in the cloud radiative forcing $\Delta$CRF $= -0.10\,\mathrm{W\,m}^{-2}$. The spatial patterns are shown in Fig. S13 in the Supplement. There is considerable local variability due to changes in the locations of clouds, but systematic effects appear to be present in Eastern Europe and East Asia. Longer simulations might allow these effects to be identified more conclusively.

## 3.4 Sensitivity study: alternative constants that underestimate $N_d$ compared to the parcel model

The changes we propose result in a small overestimation of $N_d$ relative to the parcel model in certain parts of parameter space, which could be responsible for larger changes in

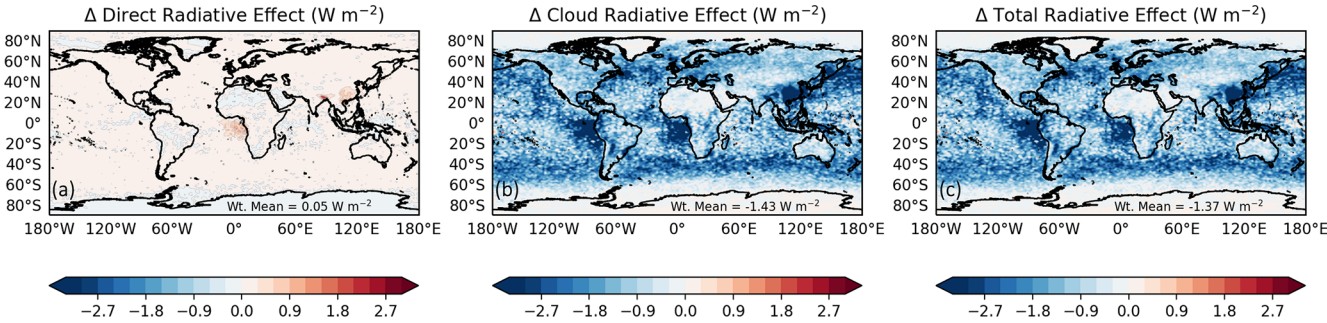

**Figure 4.** Global changes in direct radiative effects **(a)**, cloud radiative effects **(b)**, and total radiative effects **(c)** due to modifications to the ARG parameterization are shown here for 2014.

radiative effects than the changes that would be obtained if the parcel model itself were included in the climate model. Therefore, we ran another global simulation with $f = 0.70$, $g$ from Eq. (3), and $p = 1.2$. Figure S1 shows that these parameters lead to an underestimate of the $N_d$ simulated by the parcel model on average ($N_d$ is underestimated 83 % of the time) while they still improve on the default $f$ and $g$ parameters. Figure S14 in the Supplement shows a comparison of the parcel model with the ARG parameterization using these more sub-optimal constants. When we implement these revised parameters in the UM, the NMB in $N_d$ changes from $-21$ % to $-7.3$ % (compared to 2.8 % for the preferred settings). In this simulation, the change in TRE is estimated to be $-0.85 \, \mathrm{W \, m^{-2}}$ ($-0.88 \, \mathrm{W \, m^{-2}}$ for CRE). This number is much lower than the original estimate of $-1.37 \, \mathrm{W \, m^{-2}}$, but it is still substantial. The alternative $f$ and $g$ we test here may be useful to modelers who wish to mitigate the biases in the default ARG parameterization while still favoring underestimating rather than overestimating the activation fraction predicted by the parcel model.

## 4  Conclusions

We were able to improve how well the widely used ARG aerosol activation parameterization represents cloud parcel simulations of activation, focusing on how it performs for different aerosol accumulation mode widths (geometric standard deviations). By systematic comparison with a parcel model over a large parameter space, we developed updated empirical functions of mode width for the Aitken and accumulation modes for use in the parameterization. The updated ARG scheme agrees well with the parcel model over most of the parameter space, often resembling or exceeding the performance of the more sophisticated MBN parameterization. For polluted conditions, when kinetic limitations are important, we determined another adjustment to a parameter in the scheme, which extends the range of aerosol concentrations over which it reproduces parcel model results.

In most of the modal aerosol microphysics models used within climate and weather models that we are aware of,

aerosol mode width is fixed, so our suggestions amount to changing only three constant numbers, one of these subject to a simple analytic condition.

We implemented the updated ARG in the Unified Model and evaluated impacts on global cloud droplet number concentrations ($N_d$), which are substantial. Although performance of the UM degrades in the tropics, overall bias is reduced. We estimate that a $-1.43 \, \mathrm{W \, m^{-2}}$ change in cloud radiative effects would result from the changes we implement. The change in cloud radiative forcings was small but significant.

The evaluation of $N_d$ highlights the need to more carefully test and improve factors that could influence the simulated $N_d$ in future work: the aerosol and precursor emissions and microphysics, the representation of updraft speeds, and the cloud microphysics that affects wet deposition. The results especially motivate a focus on the tropics, where radiative fluxes are strongest and where the parameterization updates would worsen existing biases, at least in the UM. More detailed evaluation of $N_d$ using higher time resolution model output from a satellite simulator, and/or in situ measurements, would also be needed if tuning $N_d$ to match satellite observations were the focus of a future study. Like several other recent studies (e.g. Christensen et al., 2023), the results we show underline the importance of the baseline level of $N_d$ for climate models, as well as the change over the industrial period.

*Code and data availability.* We used the Pyrcel model (Rothenberg and Wang, 2016, 2017) version 1.3.1, which was copied to Zenodo along with the ARG modifications at https://doi.org/10.5281/zenodo.15062206 (Ghosh et al., 2025). This code includes an implementation of the ARG parameterization. The code we used to determine improved parameters and the code and data needed to reproduce all the figures in the paper is also available in the same archive. The cloud droplet concentrations derived from the MODIS satellite are publicly available at https://catalogue.ceda.ac.uk/uuid/cf97ccc802d348ec8a3b6f2995dfbbff (Grosvenor and Wood, 2018). A copy of the MODIS satellite data used in this work is available in the same archive. All atmospheric

simulations used in this work were performed using version 13.0 of the Met Office Unified Model (UM) starting from the GA7.1 configuration (Walters et al., 2019) and also included version 7.0 of JULES. All the modifications we recommend were implemented in the UKCA submodule, which is open source under a BSD-3 license. A copy of the modified versions of the UKCA code used here, along with the default version, is also available in the same Zenodo archive. Due to intellectual property copyright restrictions, we cannot provide the source code for the UM atmosphere model that hosts UKCA or the source code for JULES. The UM code is available for use under a closed license agreement. A number of research organizations and national meteorological services use the UM in collaboration with the Met Office to undertake research, produce forecasts, develop the UM code, and build and evaluate models. For further information on how to apply for a license, please contact scientific_partnerships@metoffice.gov.uk. See also http://www.metoffice.gov.uk/research/modelling-systems/unified-model (last access: 23 July 2025). UM documentation papers are accessible to registered users at https://code.metoffice.gov.uk/doc/um/latest/umdp.html (last access: 23 July 2025). JULES is freely available to any researcher for non-commercial use. Further information on requesting access and the JULES terms and conditions is accessible via http://jules-lsm.github.io/access_req/JULES_access.html (Clark et al., 2011). Rose and Cylc software were used to drive the Unified Model. The simulations were run using Rose version 2019.01.3 and Cylc version 7.8.8, which are publicly available at https://doi.org/10.5281/zenodo.3800775 (Shin et al., 2020) and https://doi.org/10.5281/zenodo.4638360 (Oliver et al., 2021), respectively. Both Rose and Cylc are available under v3 of the GNU General Public License (GPL). The full list of simulation identifiers for the simulations in this paper is given below.

- Present-day simulation, no modifications to ARG: u-dh390

- Present-day simulation, including modifications to ARG: u-dh441

- Pre-industrial simulation, no modifications to ARG: u-dh477

- Pre-industrial simulation, including modifications to ARG: u-dh478

- Present-day simulation, including modifications to ARG that tend to underestimate $N_{\mathrm{d}}$: u-dh473

- Present-day simulation, including modifications to ARG, without the suggested fix for kinetically limited activation: u-dh472

*Supplement.* The supplement related to this article is available online at [the link will be implemented upon publication]. `TS3`

*Author contributions.* PG and HG formulated the idea of the paper with important contributions from all co-authors. PG and HG set up different model configurations and designed the simulations. PG ran all the simulations. PG wrote the paper with comments and suggestions from all co-authors.

*Competing interests.* The contact author has declared that none of the authors has any competing interests.

*Acknowledgements.* This research was supported by the U.S. Air Force Life Cycle Management Center (LCMC) collaboration with Oak Ridge National Laboratory (ORNL). The computational resources on Air Force Weather HPC11 are provided by the Oak Ridge Leadership Computing Facility (OLCF) Director's Discretion Project NWP501. The OLCF at Oak Ridge National Laboratory (ORNL) is supported by the Office of Science of the U.S. Department of Energy under Contract No. DE-AC05-00OR22725. Model simulations are materials produced using Met Office software. Hamish Gordon acknowledges funding from the Department of Energy's Atmospheric System Research program and from NASA. Daniel P. Grosvenor was supported by the Met Office Hadley Centre Climate Programme funded by DSIT and Daniel P. Grosvenor acknowledges support from the Centre for Environmental Modelling And Computation (CEMAC) at the University of Leeds. This work also used the Extreme Science and Engineering Discovery Environment (XSEDE), which is supported by the National Science Foundation Grant ACI-1548562. Specifically, it used the Bridges-2 system, which is supported by the NSF Award ACI-1928147, at the Pittsburgh Supercomputing Center (PSC). We thank David O'Neal for his assistance with the installation of the UM on this system, which was made possible through the XSEDE Extended Collaborative Support Service (ECSS) program. This work also used Bridges-2 at the PSC through allocation atm200005p from the Advanced Cyberinfrastructure Coordination Ecosystem: Services & Support (ACCESS) program, which is supported by National Science Foundation grants nos. 2138259, 2138286, 2138307, 2137603, and 2138296.

*Financial support.* This research has been supported by the U.S. Department of Energy (grant no. DE-SC0022227) and the National Aeronautics and Space Administration (grant no. 80NSSC21K1344).

*Review statement.* This paper was edited by Graham Mann and reviewed by two anonymous referees.

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

**Remarks from the typesetter**

TS1 Note that changes that might alter the scientific content require editor approval. Please provide an explanation regarding this correction (0.53 to 0.97) that can be forwarded by us to the editor.

TS2 Note that changes that might alter the scientific content require editor approval. Please provide an explanation regarding this correction (0.45 to 1.95) that can be forwarded by us to the editor.

TS3 Please send a new Supplement as a *.pdf without the title, authors, correspondence author, etc. as we will generate a Supplement title page during publication (with a citation including the DOI), which will contain this information.