# Peer review of "Assessing modifications to the Abdul-Razzak & Ghan aerosol activation parameterization (version ARG2000) to improve simulated aerosol-cloud radiative effects (in UK Met Office Unified Model, version 13.0)"

_EGUsphere, 2024_

## Author Comment (AC1)

**Response to Reviewer 1**

This work presents an update to the Abdul-Razzak and Ghan (ARG) parametrization for aerosol activation. ARG is a well-established and widely used parametrization in the regional and global modelling community, so this is a very useful development with a large application potential. The paper is clearly written and the method is documented in detail, facilitating the implementation by other modelling groups. The improved parameterization is then tested and evaluated in a state-of-the-art global aerosol-climate model, demonstrating a generally improved model performance for CDNC against satellite data. The paper is definitely worth of publication in GMD. Just a few questions and remarks, which the authors may consider before publication:

We sincerely thank the reviewer for their thoughtful and constructive comments on the manuscript. We have carefully considered each comment and have addressed them in detail below, making the necessary revisions and providing additional clarifications where appropriate. Reviewer comments are in **Blue**, our responses are in **Black**, and manuscript updates are in **Red**.

1. The global model simulations performed for this study cover only 1 year (2014), so it is not possible to quantify the variability of the results. Could you briefly comment on that, also based on your previous experience with the model?

We thank the reviewer for raising this concern. We chose 2014 because this is the endpoint year for the CMIP6 project, so it makes sense to define it as 'present day' and compare the results with existing literature (included in the revised draft, see reply to the minor comments). Our model setup is very similar to the UK Earth System Model version 1.1 (UKESM1.1) discussed in Mulcahy et al. (2023). In their work, similar biases in cloud top daytime droplet number concentrations were found when compared with the Grosvenor et al. (2018) dataset for 2003 to 2015. Hence, we are confident that simulating a longer period would result in changes similar to those shown in our manuscript. However, we acknowledge that using different models (Smith et al., 2020) or versions or simulation years might result in slightly different values compared to what we obtain here.

In the revised manuscript, on line L147 we add: "Climate models that participated in the Coupled Model Intercomparison Project Phase 6 (CMIP6) also reported radiative forcing due to aerosol-cloud interactions over a period ending in the year 2014 (Forster et al., 2021)."

In line 266 we add: "The magnitude of these biases and associated cloud radiative effects might also depend on choice of model (Smith et al., 2020), or version, or the simulation year."

2. Sect. 3.2: it would be good to complement the evaluation with the RMSE in addition to the NMB, as the latter is affected by the compensation of over/underestimation.

We thank the reviewer for this great suggestion. We agree that the Normalized Mean Bias (NMB) metric can be influenced by the compensation of biases. Like RMSE, it is also asymmetric in the sense that fractional overestimation is penalized more than fractional underestimation. Therefore, to address this concern, we have calculated RMSE and also Normalized Mean Absolute Error Factor (NMAEF). We find that the RMSE becomes worse with the updated ARG (numbers below). NMAEF, which is unbiased for fractional uncertainties, decreases with the updated ARG, indicating better model performance.

In the revised manuscript, on line 240 we add:

"We calculated several metrics to quantify the model performance. The Normalized Mean Bias (NMB) is useful in that it describes whether the model overestimates or underestimates observations

overall, but positive and negative biases tend to cancel in the metric. Therefore, Root Mean Square Error (RMSE) is also commonly used to summarize the magnitude of the differences between the model and observations. However, both metrics penalize fractional overestimates more than fractional underestimates: for example, a model that overestimates observations by a factor two has a NMB of +100%, while one that underestimates observations by a factor two has a NMB of −50%. The model that overestimates has a larger absolute error but the same fractional error compared to the model that underestimates. Hence, following Gustafson and Yu (2012), we also compute the Normalized Mean Absolute Error Factor (NMAEF), which penalizes fractional overestimates and underestimates equally. It also differs from RMSE in that RMSE depends quadratically on errors while NMAEF depends on them linearly. We quote a simplified version of the NMAEF valid for positive model values and observations; the full version is given in Equation 5 of Gustafson and Yu (2012)."

In line 262 we add:

"The globally averaged RMSE increases from 64 cm$^{-3}$ to 72 cm$^{-3}$ when the ARG scheme is updated, while the global average NMAEF decreases from 0.48 to 0.39. Thus, the overall bias and fractional error decrease when we update the ARG parameterization, while the absolute error (measured by the RMSE) increases. Whether or not the performance of the model improves overall therefore depends on the priorities of the user. The magnitude of these biases and associated cloud radiative effects might also depend on choice of model (Smith et al., 2020), or version, or the simulation year."

Based on the comment of reviewer 2, we also design another sensitivity study to better understand the model performance. Please see below.

3. Would it be reasonable to also look at other variables, like liquid water path, cloud cover and precipitation? Do you expect changes in these variables with the improved ARG parametrization?

We thank the reviewer for this valuable suggestion. In Supplementary Figure S12, we show the changes in Liquid Water Path between the simulation with the updated ARG and the default ARG. The global average change is small yet systematic (3.4%). However, we find only minor changes in cloud cover and precipitation (not shown). In the revised manuscript, on line 292, we add the following:

"In Figure S12, we present the difference in liquid water path (LWP) between present-day simulations using the default ARG and the updated ARG. Our results indicate a small yet systematic increase of 1.72 g m$^{-2}$ (3.4%) in LWP with the updated ARG, concentrated in regions where cloud cover is high. We find only minor differences in precipitation and cloud cover between these simulations (not shown)."

[Figure]

Minor suggestions Line 10: you could also provide the changes of CRE and aerosol forcing in relative terms.

We thank the reviewer for their suggestion. We have provided the percentage changes of CRE and aerosol forcing in the parentheses in the abstract.

Line 40: in the conditions used, can you be more specific?

We agree with the reviewer that this statement requires further clarification. The "conditions" mentioned in the paper refer to ammonium nitrate aerosols with a hygroscopicity of 0.7 (lognormally distributed with a number mode diameter of 100 nm) and a fixed updraft velocity of 0.5 m s$^{-1}$. We have also refined the sentence structure and ensured the accuracy of the reported values.

In the revised manuscript, line 40 is now modified to the following:

"They used ammonium nitrate aerosols of hygroscopicity 0.7 (lognormally distributed with number geometric mean diameter of 100 nm) and a fixed updraft velocity of 0.5 m s$^{-1}$ to prepare that figure. Biases in the activation fraction in most schemes were small at widths of around 2.0 but increase below this. At a width of 1.6, the activation fraction in the ARG scheme is ~15% lower than the cloud parcel model, however at a width of 2.0 the fractions activated are within 10%."

Line 52: maybe add that targeting the accumulation mode is justified since this is the most relevant mode for activation?

We thank the reviewer for this suggestion. In the revised manuscript, on line 106, we add the following text:

"We vary only the mode width of the accumulation mode, as we find that it is more important to the determination of $N_d$ than than the Aitken and coarse mode widths."

Line 63: citing the original Köhler theory would be appropriate in this context.

We thank the reviewer for this suggestion and have added the appropriate reference on line 66 of the revised manuscript.

Line 90: could you add a reference to the Latin hypercube sampling method?

We thank the reviewer for pointing this out. We have added a couple of references to the Latin hypercube sampling method on line 99 of the revised manuscript.

Table 2: in the list of parameters, Aitken should appear before accumulation.

We thank the reviewer for raising this point. The table is fixed following the reviewer's suggestion

Table 3: I would sort the columns in a way that the old and new values of a given parameter appear next to each other, otherwise it is quite hard to compare them.

We thank the reviewer for their suggestion. We have fixed the Table, such that new and old values of the parameters appear next to each other.

Figure 1 caption: please use the SI units for pressure (Pa) instead of atm throughout the paper.

We thank the reviewer for raising this point. We have changed the units throughout the paper.

Figs. S8 and S9: the captions refer to "annual averages" and "January 2014", but the plot show DJF and JJA, respectively (the main text too). Please Check.

We thank the reviewer for bringing this to our attention. We confirm that Figure 3 in the main text shows the annual averages. In the supplement, we have updated the captions for Figures S9 and S10

to clarify that Figure S9 represents the DJF average, while Figure S10 corresponds to the JJA average.

*Line 253: you could mention the IPCC-AR6 ranges and the recent Bellouin et al. (2020, doi:10.1029/2019RG000660) results here, to support the statement that your RFs are reasonable.*

We thank the reviewer for this suggestion. We have mentioned the estimated radiative forcings from the sixth assessment report and the Bellouin et al. (2020) to better support our results.

In the revised manuscript, on line 311 we add the following:

"Bellouin et al. (2020) estimates the present-day aerosol effective radiative forcing to range from $-1.60$ to $-0.65\,\mathrm{Wm^{-2}}$, with a $16\%-84\%$ uncertainty range. The Sixth Assessment Report (AR6) of the Working Group I (WGI) of the Intergovernmental Panel on Climate Change (IPCC) estimates the direct radiative forcing at $-0.3\,(-0.6$ to $0.0)\,\mathrm{Wm^{-2}}$, and the cloud radiative forcing at $-1.0\,(-1.7$ to $-0.3)\,\mathrm{Wm^{-2}}$ (Forster et al., 2021) over the industrial era."

*Line 269: different accumulation mode - > different aerosol accumulation mode.*

Fixed (see Line 334).

**References**

Bellouin, N., Quaas, J., Gryspeerdt, E., Kinne, S., Stier, P., Watson-Parris, D., Boucher, O., Carslaw, K. S., Christensen, M., Daniau, A.-L., Dufresne, J.-L., Feingold, G., Fiedler, S., Forster, P., Gettelman, A., Haywood, J. M., Lohmann, U., Malavelle, F., Mauritsen, T., McCoy, D. T., Myhre, G., Mülmenstädt, J., Neubauer, D., Possner, A., Rugenstein, M., Sato, Y., Schulz, M., Schwartz, S. E., Sourdeval, O., Storelvmo, T., Toll, V., Winker, D., and Stevens, B. (2020). Bounding global aerosol radiative forcing of climate change. *Reviews of Geophysics*, 58(1):e2019RG000660. e2019RG000660 10.1029/2019RG000660.

Forster, P., Storelvmo, T., Armour, K., Collins, W., Dufresne, J.-L., Frame, D., Lunt, D. J., Mauritsen, T., Palmer, M. D., Watanabe, M., Wild, M., and Zhang, X. (2021). The earth's energy budget, climate feedbacks, and climate sensitivity. In Masson-Delmotte, V., Zhai, P., Pirani, A., Connors, S. L., Péan, C., Berger, S., Caud, N., Chen, Y., Goldfarb, L., Gomis, M. I., Huang, M., Leitzell, K., Lonnoy, E., Matthews, J. B. R., Maycock, T. K., Waterfield, T., Yelekçi, O., Yu, R., and Zhou, B., editors, *Climate Change 2021: The Physical Science Basis. Contribution of Working Group I to the Sixth Assessment Report of the Intergovernmental Panel on Climate Change*, pages 923–1054. Cambridge University Press, Cambridge, United Kingdom and New York, NY, USA.

Grosvenor, D. P., Sourdeval, O., Zuidema, P., Ackerman, A., Alexandrov, M. D., Bennartz, R., Boers, R., Cairns, B., Chiu, J. C., Christensen, M., Deneke, H., Diamond, M., Feingold, G., Fridlind, A., Hünerbein, A., Knist, C., Kollias, P., Marshak, A., McCoy, D., Merk, D., Painemal, D., Rausch, J., Rosenfeld, D., Russchenberg, H., Seifert, P., Sinclair, K., Stier, P., van Diedenhoven, B., Wendisch, M., Werner, F., Wood, R., Zhang, Z., and Quaas, J. (2018). Remote sensing of droplet number concentration in warm clouds: A review of the current state of knowledge and perspectives. *Reviews of Geophysics*, 56(2):409–453.

Gustafson, W. I. and Yu, S. (2012). Generalized approach for using unbiased symmetric metrics with negative values: normalized mean bias factor and normalized mean absolute error factor. *Atmospheric Science Letters*, 13.

Mulcahy, J. P., Jones, C. G., Rumbold, S. T., Kuhlbrodt, T., Dittus, A. J., Blockley, E. W., Yool, A.,

Walton, J., Hardacre, C., Andrews, T., Bodas-Salcedo, A., Stringer, M., de Mora, L., Harris, P., Hill, R., Kelley, D., Robertson, E., and Tang, Y. (2023). Ukesm1.1: development and evaluation of an updated configuration of the uk earth system model. *Geoscientific Model Development*, 16(6):1569–1600.

Smith, C. J., Kramer, R. J., Myhre, G., Alterskjær, K., Collins, W., Sima, A., Boucher, O., Dufresne, J.-L., Nabat, P., Michou, M., Yukimoto, S., Cole, J., Paynter, D., Shiogama, H., O'Connor, F. M., Robertson, E., Wiltshire, A., Andrews, T., Hannay, C., Miller, R., Nazarenko, L., Kirkevåg, A., Olivié, D., Fiedler, S., Lewinschal, A., Mackallah, C., Dix, M., Pincus, R., and Forster, P. M. (2020). Effective radiative forcing and adjustments in cmip6 models. *Atmospheric Chemistry and Physics*, 20(16):9591–9618.

---

## Author Comment (AC2)

**Response to Reviewer 2**

This paper used a parcel model to explore the parameter space and optimize the performance of the widely used Abdul-Razzak & Ghan aerosol activation scheme (ARG2000). The research question is well-defined and the comparison with other schemes and parcel model results is comprehensive. This paper addresses an important topic in aerosol-cloud interaction. However, the evaluation within the global model context could benefit from additional rigor. I recommend publication in GMD after addressing the comments below.

We sincerely thank the reviewer for their thoughtful and constructive comments on the manuscript. We have carefully considered each comment and have addressed them in detail below, making the necessary revisions and providing additional clarifications where appropriate. Reviewer comments are in **Blue**, our responses are in **Black**, and manuscript updates are in **Red**.

General Comment on Nd Comparison: While the comparison between model output Nd and MODIS Nd is interesting, it may be more complex due to the various differences between the model and satellite data. These differences make it challenging to determine whether the bias in the model with the old scheme is 'underestimated' or 'overestimated,' and consequently, whether the reported 'improvements' or 'degradations' are fully reliable. I believe the comparison using the MODIS simulator results might be more appropriate than using the model's Nd output.

We thank the reviewer for their suggestions. We agree that model-observation comparisons are inherently challenging, and it is difficult to definitively determine whether the updated parameterization performs better or worse due to the large uncertainties present in both the model and the observations. We have explored several approaches to address these shortcomings in the revised manuscript. First, we provide a better description of the cloud top $N_d$ which attempts to mimic the way a satellite would see the cloud top. Second, based on Reviewer 1's suggestions, we calculate the error and report the magnitude to better evaluate the model performance. Finally, We also attempt to crudely match the Grosvenor et al. (2018) filtering criteria. Kindly see replies to the comments below.

The current study focuses mainly on updating the ARG parameterization, with the global model-observation comparison only intended to add context to the results. Although comparison with a MODIS satellite simulator could provide valuable insights, it is unclear whether it would be an improvement. The model simulates $N_d$. To use a satellite simulator such as COSP, we would need to use the effective radius and optical depth predicted by the satellite simulator to recalculate $N_d$ using a similar algorithm to the satellite data. These variables are derived from the simulated $N_d$ and liquid water content, and are therefore affected by assumptions about the droplet size distribution in the model's radiation scheme which do not impact the $N_d$ we currently show. $N_d$ predicted in this way is therefore less directly related to the ARG algorithm, and involves more modeling assumptions. It might be more closely related to the cloud radiative effect, but it would likely not be the same as the $N_d$ predicted by the ARG algorithm.

In addition, to avoid excessive disk space usage we write out monthly mean $N_d$ from our model, but if we wrote out monthly mean effective radius and optical depth, then calculated $N_d$, the $N_d$ would be biased because it depends non-linearly on effective radius and optical depth. It would still be feasible to write these out and calculate $N_d$ (say) every three hours in principle, and we agree that doing the comparison in multiple different ways would be interesting, even if there is no clear right answer.

We also think based on other work published and in progress that many of the biases in $N_d$ we can identify are unrelated to the activation parameterization, but are instead due to aerosol concentration or updraft speed biases. Therefore a more careful comparison of $N_d$ with observations would be of limited

value without a corresponding evaluation of the aerosols and updraft speeds. All in all this would be a major undertaking we prefer to leave for a dedicated future study. As we comment in the text, "The biases likely originate mainly in aerosol modeling unrelated to the activation parameterization, or in the sub-grid updraft speed. The satellite retrievals used for the evaluation are also uncertain, and the comparison is imprecise due to the representativeness uncertainty". That said, we still feel that it is of value to set the improvements we present to the activation algorithm in the context of likely model biases and to draw the readers' attention here to the probable scale of the problems that must be addressed in this future work.

L210: It would be helpful to provide more details about how the model's $N_d$ is calculated. Specifically, how is the liquid cloud-top defined in the model, and what methodology is used to derive Nd from this definition?

We thank the reviewer for their comment. We agree that it is important to describe in detail how cloud top droplet concentrations ($N_d$) are calculated in the model. The model diagnostic we use (which has existed in the model for many years but we are not aware of a reference) attempts to mimic the way a satellite would see the cloud top. In the revised manuscript, on line 231 we add the following:

"The cloud top is defined as the highest model level with non-zero liquid cloud fraction. If the cloud fraction is 100%, then $N_d$ at that level is taken as the cloud top $N_d$. If not, the model-level-weighted $N_d$ is calculated by summing over contributions from different model levels here indexed by $i$, as:

$$N_d = \frac{\sum_{i=1}^{n} N_d(i) \cdot cf(i) \cdot Pr(i)}{\sum_{i=1}^{n} cf(i) \cdot Pr(i)} \tag{1}$$

Here $N_d(i)$ is the droplet number concentration at the $i$-th model level. $cf(i)$ is the liquid cloud fraction at the $i$-th model level. $Pr(i)$ is the probability that a photon leaving the $i$-th layer can escape to space without encountering a cloud in the layers above. This is calculated based on the cloud fraction in the layer above the $i$-th layer. $n$ is the total number of model levels. This equation is calculated only for the daytime. The cloud-top $N_d$ is calculated in each timestep, and the monthly average is written out as an output diagnostic."

L215: The MODIS Nd estimates from Grosvenor et al. (2018) involve several thresholds to sample clouds. I would recommend clarifying whether similar thresholds are applied to your model results. If not, it would be valuable to discuss how the sampling differences due to post-processing thresholds and simulator assumptions may impact the comparison presented in Figure 3. Addressing these differences will enhance the clarity and reliability of the evaluation.

We thank the reviewer for their comment. We agree that the Grosvenor et al. (2018) dataset (GW2018) has several assumptions and thresholds to calculate $N_d$. We are able to apply most (but not all) of the thresholds while calculating the modeled $N_d$. The GW2018 dataset has a daily frequency, while the model provides monthly averages. First, we calculate the monthly mean GW2018 $N_d$ using all available data points. Then we regrid it to the model gridpoints and filter out the model data (only liquid clouds) where we do not have any valid GW2018 retrievals. The GW2018 dataset includes retrievals only where the $1° \times 1°$ cloud fraction is at least 80%. Applying this threshold to the model output in Figure 3 is challenging due to the use of monthly-averaged cloud fraction.

In response to the comment, we perform a new sensitivity study. Since achieving precise filtering would require 3-hourly model outputs, we adopt an approximate approach to exclude gridboxes with low cloud fraction. Specifically, for each month, we filter out the bottom 25th percentile of weights

(the denominator in the daytime cloud-top $N_d$ equation) in the simulations and mask the corresponding MODIS data in those gridboxes. This ensures that we sample only gridboxes with relatively higher cloud fractions. In Supplement Figure S11, we show the model-observation comparison with this filtering technique. We also report the Normalized Mean Bias (NMB), Root Mean Square Error (RMSE), and Normalized Mean Absolute Error Factor (NMAEF). In the revised manuscript, in line 268 we add the following:

"The Grosvenor et al. (2018) dataset has several assumptions and thresholds to calculate $N_d$. An important assumption is a minimum of 80% cloud cover in a $1° \times 1°$ gridbox. Our model diagnostics are written out as monthly averages, to avoid the high cost in disk space associated with output at higher time resolution, and hence it is difficult to apply exactly the same threshold. We conduct a sensitivity study by filtering out the bottom 25th percentile of weights (the denominator in Equation 6) and their corresponding simulated $N_d$ values, and mask out the same grid cells from the monthly-averaged MODIS dataset. This filtering ensures that we select gridboxes which has relative higher cloud fractions on average in the model. In Figure S8, we show the simulated and observed $N_d$ with this filter in place. We find that the global NMB change with the updated ARG is similar to the unfiltered case (a change from -22% to 0.1% compared to the previous change from -21% to 2.8%). The changes to the RMSE and NMAEF with the filter are also small (RMSE increases from 59 cm$^{-3}$ to 65 cm$^{-3}$ compared to the previous change from 64 cm$^{-3}$ to 72 cm$^{-3}$). While our investigation remains approximate, there is no strong evidence from this test that our results would be substantially changed if we were able to match the simulations to satellite data more precisely."

[Figure]

Section 3.4: The conclusion drawn in this section is somewhat unclear. It would be helpful to either explicitly state the implications of the findings or, if they are not central to the paper's argument, consider removing or rewording this part. This would enhance the clarity and focus of the discussion.

We thank the reviewer for the suggestion. We agree that the motivation for Section 3.4 was not very clear. Hence, in the revised manuscript, on line 329 we add the following:

"The alternative $f$ and $g$ we test here may be useful to modelers who wish to mitigate the biases in the default ARG parameterization while still favoring underestimating rather than overestimating the activation fraction predicted by the parcel model."

 To improve clarity, it would be helpful to show the formula of non-dimensional variables $\eta$ and $\zeta$. This would allow readers to better understand the discussion around kinetic limitations and how these variables related to Nd and w.

We thank the reviewer for this suggestion. We agree that a better description of $\eta$ and $\zeta$ would help the readers better understand our updates to the ARG scheme. Hence, following the suggestions of the reviewer, we added the expression for $\eta$ and $\zeta$, and in the revised manuscript, in line 84 we have added a brief description.

"$\zeta$ and $\eta_i$ are given by: :

$$\zeta = \frac{2A}{3}\left(\frac{\alpha w}{G}\right)^{\frac{1}{2}}$$
$$\eta_i = \frac{2\left(\frac{\alpha w}{G}\right)^{\frac{3}{2}}}{\gamma^* N_i}.$$

Here $A$ is the Kelvin coefficient (a function of temperature), $w$ is the updraft velocity, $\alpha$ is a thermodynamic term (function of temperature) that relates the updrafts to the tendency for water vapor to condense as it cools, $N_i$ is the aerosol number concentration in mode $i$, $\gamma^*$ follows from the thermodynamics of rising moist air with assumptions listed in Pruppacher and Klett (2010), and $G$ is the growth coefficient which depends on the diffusivity of water vapor in air and on the thermal conductivity of the air. The ratio of $\zeta$ and $\eta_i$ is proportional to the ratio of $N_i$ and $w$. Since none of these parameters depend of the mode width, Abdul-Razzak and Ghan (2000) parameterized the dependence of $S_{max}$ on $\sigma$ by using two additional parameters, $f$ and $g$."

 Could you clarify the nudging relaxation time scale?

We thank the reviewer for this comment. We agree that the choice of relaxation parameter is important. In this work, we set the relaxation parameter to $1/6\ \mathrm{h}^{-1}$. In the revised manuscript, on line 141 we add the following:

"The choice of the nudging relaxation parameter is important; too small a value makes nudging ineffective, while too large a value can destabilize the model. Following Telford et al. (2008), we use the relaxation parameter of $1/6\ \mathrm{h}^{-1}$, corresponding to the time spacing of the ERA5 data."

 'AAF' should be changed into 'TAF'.

Changed (see line 189)

 the modification to the constant 'p' does not fix the underlying problem but alleviates it. It would be insightful to discuss why the improvement appears to be limited and consider other possible explanations or avenues for further improvement.

We thank the reviewer for their suggestion. We agree that more discussion is needed to better understand the limitations of the fix we proposed in the manuscript. Some activation parameterizations are able to treat kinetic limitations in a more principled way (Nenes and Seinfeld, 2003; Morales Betancourt and Nenes, 2014; Ming et al., 2006). However, some empirical relationships are still involved, for example to define the regime where kinetic limitations become dominant (Nenes and Seinfeld, 2003). Our approach maintains the simplicity of the ARG algorithm, in which no iterations or bisections are needed, but its disadvantage is that the kinetic growth is not treated in a physically motivated way: this is the reason for the persistence of poor performance in extreme cases.

If we tried to add more physics we would probably end up with something similar to the Nenes and Seinfeld (2003); Morales Betancourt and Nenes (2014); Ming et al. (2006) algorithms. Alternatively, of

course, one could also guarantee fidelity to the parcel model using machine learning-based emulators, for example those of Silva et al. (2021). This would require larger changes to the code base of a climate model to implement, and may reduce the ease with which the behavior of the scheme can be interpreted.

More sophistication, for example considering hydration of aerosols and accounting for giant CCN using expressions based on first principles, could also improve model performance.

In the revised manuscript, on line 219 we add the following:

"Unlike our modification, the approaches of Nenes and Seinfeld (2003); Ming et al. (2006) have physically motivated treatments of kinetic droplet growth. However, introducing additional physics into the ARG scheme would likely lead to a formulation similar to those algorithms, compromising the simplicity. Alternatively, of course, one could also guarantee fidelity to the parcel model using machine learning-based emulators, for example, those of Silva et al. (2021). This would require larger changes to the code base of a climate model to implement and may reduce the ease with which the behavior of the scheme can be interpreted."

L241: 'Figure S12' should be changed into 'Figure 4'.

Changed (see line 300)

**References**

Abdul-Razzak, H. and Ghan, S. (2000). A parameterization of aerosol activation: 2. multiple aerosol types. *Journal of Geophysical Research*, 105:6837–6844.

Grosvenor, D. P., Sourdeval, O., Zuidema, P., Ackerman, A., Alexandrov, M. D., Bennartz, R., Boers, R., Cairns, B., Chiu, J. C., Christensen, M., Deneke, H., Diamond, M., Feingold, G., Fridlind, A., Hünerbein, A., Knist, C., Kollias, P., Marshak, A., McCoy, D., Merk, D., Painemal, D., Rausch, J., Rosenfeld, D., Russchenberg, H., Seifert, P., Sinclair, K., Stier, P., van Diedenhoven, B., Wendisch, M., Werner, F., Wood, R., Zhang, Z., and Quaas, J. (2018). Remote sensing of droplet number concentration in warm clouds: A review of the current state of knowledge and perspectives. *Reviews of Geophysics*, 56(2):409–453.

Ming, Y., Ramaswamy, V., Donner, L. J., and Phillips, V. T. J. (2006). A new parameterization of cloud droplet activation applicable to general circulation models. *Journal of the Atmospheric Sciences*, 63(4):1348 – 1356.

Morales Betancourt, R. and Nenes, A. (2014). Droplet activation parameterization: the population-splitting concept revisited. *Geoscientific Model Development*, 7(5):2345–2357.

Nenes, A. and Seinfeld, J. H. (2003). Parameterization of cloud droplet formation in global climate models. *Journal of Geophysical Research: Atmospheres*, 108(D14).

Pruppacher, H. and Klett, J. (2010). *Microphysics of Clouds and Precipitation*, volume 18.

Silva, S. J., Ma, P.-L., Hardin, J. C., and Rothenberg, D. (2021). Physically regularized machine learning emulators of aerosol activation. *Geoscientific Model Development*, 14(5):3067–3077.

Telford, P. J., Braesicke, P., Morgenstern, O., and Pyle, J. A. (2008). Technical note: Description and assessment of a nudged version of the new dynamics unified model. *Atmospheric Chemistry and Physics*, 8(6):1701–1712.

---

## Author Response (AR2)

Dear Graham,

Thank you for your positive feedback on our manuscript. Please find below our detailed responses to your comments. Your original comments are shown in blue, and our replies are in black.

1) Manuscript title, lines 1-4. (title needs further amending).

The title was revised following the initial Topical Editor review, primarily to ensure adherence to the GMD policy, for then having an indicator "ARG2000" to denote a specific version of the activation parameterization, with also the version number for the host UK Met Office Unified model (v13.0). However, further slight amendment is needed, as although technically not grammatically incorrect, the statement made there does not actually summarise what the manuscript finds. The manuscript type is model evaluation paper, and it's this specifics of the evaluation that needs to be stated within the title there. Specifically, please change "Modifying the. . . ." Instead to "Assessing modifications to the. . . ". The grammar of the statement also is not great, and please change "impacts simulated cloud radiative effects" instead to "to improve simulated aerosol-cloud radiative effects". The words "shown" and "the" should also be deleted from the parentheses to keep the title concise.

We thank you for this suggestion. Based on your suggestion the title is changed to:

"Assessing modifications to the Abdul-Razzak & Ghan aerosol activation parameterization (version ARG2000) to improve simulated aerosol-cloud radiative effects (in UK Met Office Unified Model, version 13.0)"

2) Section 3.2, lines 238

This equation has been added following Reviewer 2's comments (see general comments above), but the formalism here for the summation indices needs to be improved. With this being a specialist model development journal the formalism ought to be more precise, so please use sub-script i rather than i in parentheses for each term.

We thank you for the suggestion. We have fixed the subscripts based on your suggestions.

3) Section 3.2, lines 238-241

Further to comment 2, the variable names here "cf" and "Pr" should be presented with just 1 main symbol to denote them, with subscripts for indicating any secondary specifications from the primary quantity denoted. There are 2 instances here where that's not the case, the "cf" for "cloud fraction" and the "Pr" for probability. In both cases there is a recognised roman letter that tends to be used within equations, (capital F for cloud fraction, and capital P for probability), and then the secondary aspects should only be communicated via sub-scripts. Since the cloud fraction is clarified to be liquid cloud fraction, I suggest sub-script lc for liquid cloud, with a comma then to the sub-script I for the summation (i.e. F subscript "cl,I"). Although Pr(i) represents a probability for photon transfer, suggest to denote simply as P subscript I to focus the main attention on the other variables, these being the primary quantities to consider. Please also amend the text on lines 240-241 accordingly.

Based on your suggestions, we have modified the equation and the associated text. See line 235 of the revised manuscript. In the tracked changes document, the old equation is shown in red, and the updated equation is shown in blue.

4) Section 3.2, line 267-268

Be more specific here, re: "in the cloudiest regions which most affect the global average" The North Pacific has the region of persistent marine stratocumulus off the coast of the Pacific Northwestern US, and I guess that will be one of the "poleward of 30N cloudiest regions" you mean there, right? Are there similarly 1 (or possibly 2) other regions of persistent low cloud that could be stated there (e.g. within the North Atlantic?)

Thank you for this helpful suggestion. Indeed, the region of persistent marine stratocumulus off the coast of the Pacific Northwestern United States is one of the cloudiest areas poleward of 30°N to which we refer. In addition, the North Atlantic, particularly the region off the coast of Western Europe and the northwestern Atlantic near Newfoundland, represents another key area of persistent low-level cloud cover. We have revised the text to explicitly mention these two regions for greater clarity. In line 260 of the revised manuscript, we add the following:

"Poleward of 30° latitude, in the cloudiest regions (such as the coast of the Pacific Northwestern United States, the coast of western Europe, or Newfoundland in the northwestern Atlantic, for example) which most affect the global average, our updated ARG parameterization leads to an improvement in the NMB..."

5) Section 3.2, lines 279-280

The "We conduct a sensitivity study by filtering out..." needs to be re-worded – since this is 1 model run, the "study" is too big a word. I guess you mean "sensitivity experiment" here, right?. With this sentivity added from one of Reviewer 2's comments (RC2-MC1), it's important to mention this is to assess the temporal sampling. Suggest to re-word as "To assess temporary sampling in the monthly means, we carried out a sensitivity experiment, filtering out. . . . ". 6) Section 3.2, line 285-286

Thank you for the suggestion. We agree that 'sensitivity experiment' is a more appropriate term in this context. However, we believe that 'temporal sampling' refers to the frequency of data averaging over time. Since we consistently use monthly averaged model data, this term does not apply here. Instead, we are evaluating the sensitivity of $N_d$ to the filtering of the liquid cloud fraction. Accordingly, on line 274 of the revised manuscript, we have revised the text to read:

"To understand the impact of this cloud fraction threshold on the temporally averaged $N_d$, we design a sensitivity experiment by filtering out the bottom 25th percentile of weights within each month ..."

Further to comment 5, with RC2's comments, re-word "While our investigation remains approximate.." to be clear this is an alternate methodology: "Whist this alternate methodology is approximate, it provides an indication of the sensitivity to the temporal sampling, and the results indicate only a modest change (12.5%), not changing the results substantially."

We thank you for this suggestion. We have reworded the sentence to clarify that this represents an alternative methodology. However, we respectfully disagree with the characterization of the changes as "modest". Based on the updated NMB and RMSE values, we believe that the changes are minimal. Accordingly, on line 280 of the revised manuscript, we have added the following:

"Although this alternative methodology remains approximate, it illustrates the sensitivity of the cloud fraction threshold to $N_d$. However, we did not observe significant changes in the results. Therefore, ..."

7) Section 3.2, line 289: "The spatial pattern of the bias in Figure 3 are consistent with. . . " "The spatial pattern of the CDN biases (Figure 3), are consistent with. . . .."

Line 285 of the revised manuscript is changed according to your suggestion.

8) Section 3.2, line 290: "the biases likely originate in aerosol modelling unrelated to the activation", The word "likely" is too strong an association here, and this is not evidenced in the manuscript. The biases could also potentially originate in the cloud processes, I'd argue with equal probability. Please change to ". . . the biases are quite likely unrelated to the activation parameterization, i.e. some more general bias (e.g. in either aerosol or cloud)"

We thank you for this suggestion. In line 286 of the revised manuscript, we add the following:

"The biases quite likely originate from aerosol modeling, cloud processes or updraft speeds, unrelated to the activation parameterization."

9) Section 3.2, lines 292-293: "it is still helpful to understand the impact of our improved functions"

As mentioned in the general comments, please change all instances of "our results", "our method" etc. (too colloquial), needs to be informal/neutral language in peer-reviewed journal article.

The wording "improved functions" also pre-judges the outcome of the results, and although the revised title states this is the aim, this should also be moderated to remain "the new parameterisation" or "the revised parameter settings" or parameter ranges etc.

Thank you for the suggestion. We have carefully revised the manuscript to replace instances of 'our results', 'our method', and similar phrases with more neutral and formal language appropriate for GMD. Also in line 289 of the revised manuscript, we have changed 'improved functions' to 'updated functions'. We do not think that parameter is the right word for $f$ and $g$, since these are functions of the mode width.

10) Section 3.2: line 322, "direct radiative forcing" –> "aerosol-radiation interaction radiative effect" and "cloud radiative forcing" –> "aerosol-cloud interaction radiative effect" (or "effective radiative forcing" if appropriate). The preceding sentence to this (on lines 320-321) is clear in referring to aerosol radiative forcings, and there also specifies with the IPCC report terminology "effective radiative forcing". This follow-on sentence needs also to use the approved terminology (direct –> "aerosol-radiation interaction", and indirect or cloud radiative effect –> "aerosol-cloud interaction" radiative effect. And also the instances of "direct" and "indirect" on line 324.

Thank you for raising this concern. Based on your suggestions, the text in line 317 of the revised manuscript now reads:

"The Sixth Assessment Report (AR6) of the Working Group I (WGI) of the Intergovernmental Panel on Climate Change (IPCC) estimates the effective radiative forcing due to aerosol-radiation interaction at $-0.3$ ($-0.6$ to $0.0$) Wm$^{-2}$, and the effective radiative forcing due to aerosol-cloud interaction at $-1.0$ ($-1.7$ to $-0.3$) Wm$^{-2}$ (Forster et al., 2021) over the industrial era (1750-2014). In the nudged simulations, we may not capture all adjustments (Zhang et al., 2014), and hence do not use the term 'Effective Radiative Forcing', but the simulated aerosol-radiation and aerosol-cloud radiative forcings are reasonable at $-0.20$ and $-0.95$ Wm$^{-2}$ respectively."

11) Section 3.2, line 350 to 364, Further to the above please change the several instances of "our" in this paragraph to the impersonal "the" (line 350, line 356, line 358, line 362). Change also "Like several other recent studies (for example Christensen et al., 2023), our results underline the importance of ..." to "Several other studies (e.g. Christensen et al., 2023) have highlighted the importance of. . ."

We thank you for the suggestion. We have replaced "our" with "the" in all four instances as recommended. However, regarding the final sentence, we believe it is important to emphasize that our findings are in line with previous studies, rather than implying that previous studies support our work. Therefore, we have retained the original sentence structure at line 359 of the revised manuscript.

For the "Our proposed changes are extremely simple", suggest simply to delete that, beginning the paragraph "In all modal aerosol microphysics models we are aware of".

Sentence deleted according to your suggestion.

However that sentence needs to be clarified to climate model aerosol schemes. There are certainly modal aerosol microphysics schemes in regional air quality models that are generalisable to variable mode widths (e.g. see Figure 4 of Whitby, E. et al., 2002, and see also Whitby E. et al., 1991; Seigneur et al., 1986).

We thank you for bringing this to our notice. Based on your suggestion, in Line 348 of the revised manuscript, we have updated the text to:

"In most of the modal aerosol microphysics models used within climate and weather models that we are aware of..."

12) Caption to Figure S8: Re-word the start of the penultimate sentence, currently worded as "To prepare this figure from that diagnostic, for each month we find...". The text explaining the method seems slightly oblique also. Suggest to replace this last sentence with "For this Figure, the model CDN is diagnosed with different temporal sampling, omitting gridboxes with less than 25% liquid cloud fraction (see section 3.2)."

We thank you for your suggestion. However, we do not omit the gridboxes with less than 25% cloud fraction. As mentioned in the main text, we omit only those gridboxes where the 'weight' (as defined in Equation 2) is lower than the 25th percentile of weights in all the gridboxes. Hence, mentioning cloud fraction would be inappropriate. However, we agree that the last line can be improved. Hence, we update the figure caption in the revised manuscript as follows:

"For this figure, we use this monthly diagnostic to identify the 25th percentile of the weight across the entire domain and exclude grid boxes with weights below this threshold."

**References**

Forster, P., Storelvmo, T., Armour, K., Collins, W., Dufresne, J.-L., Frame, D., Lunt, D. J., Mauritsen, T., Palmer, M. D., Watanabe, M., Wild, M., and Zhang, X. (2021). The earth's energy budget, climate feedbacks, and climate sensitivity. In Masson-Delmotte, V., Zhai, P., Pirani, A., Connors, S. L., Péan, C., Berger, S., Caud, N., Chen, Y., Goldfarb, L., Gomis, M. I., Huang, M., Leitzell, K., Lonnoy, E., Matthews, J. B. R., Maycock, T. K., Waterfield, T., Yelekçi, O., Yu, R., and Zhou, B., editors, *Climate Change 2021: The Physical Science Basis. Contribution of Working Group I to the Sixth Assessment Report of the Intergovernmental Panel on Climate Change*, pages 923–1054. Cambridge University Press, Cambridge, United Kingdom and New York, NY, USA.

Zhang, K., Wan, H., Liu, X., Ghan, S. J., Kooperman, G. J., Ma, P.-L., Rasch, P. J., Neubauer, D., and Lohmann, U. (2014). Technical note: On the use of nudging for aerosol–climate model intercomparison studies. *Atmospheric Chemistry and Physics*, 14(16):8631–8645.

---

## Author Response (AR3)

Dear Graham,

Thank you very much for the acceptance of our manuscript. We sincerely appreciate your careful review, and the constructive feedback provided throughout the editorial process.

Please find below our responses to the final grammar-level corrections. Your original comments are shown in blue, and our replies follow in black.

GLC1) Slight further grammar-level revision of the title of the manuscript

This final grammar-level revision change to the title is to have the parenthesis be restricted only to the revision number of the UM, i.e. to change: "to improve simulated aerosol-cloud radiative effects (in UK Met Office Unified Model, version 13.0)" instead to "to improve simulated aerosol-cloud radiative effects in the UK Met Office Unified Model (UM version 13.0)"

This change has been implemented in the manuscript title.

GLC2) Line 282 (of TC-MS)   "gridboxes which has"   "gridboxes which have" (plural)

This correction has been made.

GLC3) Remaining instance of colloquialism here, "we may not capture….".

Please change: "In the nudged simulations, we may not capture all adjustments (Zhang et al., 2014)…." Instead to "The nudged simulations may not capture all adjustments (Zhang et al., 2014)…."

This change has been implemented as suggested.

GLC4) Caption to Figure S8 -   Please change "for this figure" to "for this Figure" (if the word "figure" refers to Figure within a manuscript, the 1st letter is capitalised)

This correction has been made in the caption of Figure S8.